# Linking electrocatalytic turnover to elementary step rates in hydrocarbon fuel oxidation

Alexander J. Zielinski [1], Christine Lucky [2] & Marcel Schreier [1,2] ✉

Maximizing steady-state turnover rates is a central goal in electrocatalysis research, but improving one reaction step often impedes others. Navigating these trade-offs requires methods that systematically reveal how a single parameter change affects all key steps of a reaction mechanism. Here, we use electrochemical mass spectrometry to determine the potential-dependent rates of each principal step in propane oxidation on Pt and directly relate them to the steady-state turnover rate. Our analysis reveals that low steady-state activity arises from a mismatch between the optimal potentials for adsorption, conversion, and *CO oxidation. By applying alternating potentials to individually optimize adsorption and oxidation, we overcome this limitation and achieve rates exceeding those under constant-potential operation. This step-resolved approach clarifies how individual processes interact to govern overall activity and provides a framework for the rational design of electrocatalysts carrying out complex reactions at steady-state.

Electrocatalytic reactions proceed through a series of sequential steps, and the overall reaction rate is governed by the extent to which the rates of these individual steps overlap. In many cases, efforts to optimize electrocatalytic reactions take an empirical approach, focusing on tuning the binding strength of intermediates or the composition of the electrolyte[1,2]. However, changes intended to accelerate the rate-limiting step invariably affect all other steps, making this approach akin to hitting a moving target. Rational design of electrocatalytic systems therefore requires insight into how modifications to catalyst and interfacial properties impact each principal step of the reaction independently.

In this report, we establish a quantitative connection between the steady-state rate of electrocatalytic propane oxidation and the rates of its principal steps as a function of electrode potential. Using electrochemical mass spectrometry (EC-MS), we show that the overall oxidation rate (herein called 'turnover') is highest within the potential range where the rate of adsorption, adsorbate conversion, and oxidation overlap to the maximum extent. However, we also found that at this maximum turnover rate, each individual step proceeds more slowly than it would when carried out at its respective optimal

potential. This finding highlights a fundamental limitation of alkane oxidation reactions and suggests that substantially higher oxidation rates could be achieved if these steps could be made to coexist more effectively.

Alkane oxidation processes are poised to play a critical role in future energy systems. Carbon-based fuels are expected to remain in use due to their high energy density and widespread availablilty[3–7]. Moreover, such fuels are likely to be part of a sustainable energy economy. This is reflected in the significant research efforts devoted to their sustainable synthesis via electrocatalytic $CO_2$ reduction and catalytic biomass conversion[8–26]. Yet, aside from combustion, we lack efficient technologies for recovering the energy stored in these fuels. The development of hydrocarbon energy conversion technologies that surpass combustion in efficiency, reduce pollutants such as $NO_x$, and simplify system design represents a key opportunity in the transition to a more efficient energy economy. Fuel cells capable of generating electricity from electrocatalytic hydrocarbon oxidation offer a promising path toward meeting these goals (Fig. 1a)[27].

Low-temperature direct hydrocarbon fuel cells (LT-DHFC) have garnered interest in past and present-day research[28–39]. Nevertheless,

[1]Department of Chemistry, University of Wisconsin–Madison, Madison, WI 53706, USA. [2]Department of Chemical and Biological Engineering, University of Wisconsin–Madison, Madison, WI 53706, USA. ✉e-mail: mschreier2@wisc.edu

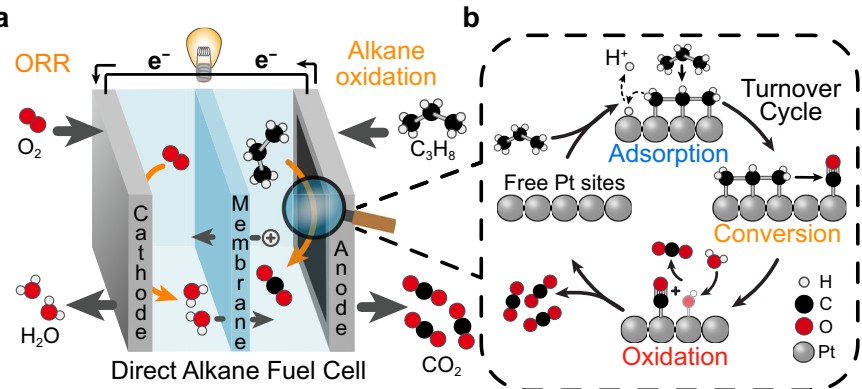

**Fig. 1 | The principal steps of electrocatalytic alkane oxidation. a** Illustration of a direct propane fuel cell. Propane undergoes total oxidation to carbon dioxide at the anode and oxygen is reduced to water at the cathode. **b** Illustration depicting the three principal reaction steps of electrocatalytic alkane oxidative turnover. Only one of several possible multi-carbon adsorbates is shown for simplicity.

their performance is poor compared to hydrogen fuel cells[40–50]. Similarly, fuel cells operating on oxygenates such as alcohols suffer from low efficiency and incomplete oxidation of alkyl chains[51–69]. A key limitation across these systems is the inadequate understanding of the mechanistic bottlenecks in steady-state electrocatalytic alkyl chain oxidation. To address this gap, the fundamental study presented herein investigates the electrocatalytic oxidation of alkanes, using propane as a model alkane, and systematically relates the steady-state oxidation rate to the potential-dependent kinetics of its principal steps.

Alkane total oxidation on Pt-based electrocatalysts is generally understood to proceed through three principal steps (Fig. 1b): (1) dissociative adsorption of alkanes onto the catalyst surface ("adsorption"), (2) fragmentation and conversion of surface bound multi-carbon adsorbates to carbon monoxide ("conversion"), (3) oxidation of surface-bound carbon monoxide (*CO) to $CO_2$ ("oxidation")[47,70–76]. Steady-state oxidation requires that all three steps occur concurrently[70].

To date, there is no consensus on which step limits the overall rate of alkane oxidation under steady-state conditions[45,47,70,72–74,76–80]. We propose that this uncertainty arises because the rate-limiting step can shift with applied potential[73,74,76]. Directly measuring the rates of individual steps would provide crucial mechanistic insight, but previous voltammetric methods have been limited in their ability to isolate step-specific contributions, instead integrating the oxidative currents from all reaction steps into a single, composite signal[70,71,76,77,81]. In contrast, we employed EC-MS to directly quantify $CO_2$ evolution from propane oxidation steps, allowing us to distinguish between product forming reactions and surface-bound oxidative processes[82–85]. By applying carefully designed electrode potential programs, we were able to map the potential dependence of each principal step and assess its contribution to the overall reaction rate[71,74,76,77,81,86–89].

Our results reveal that the steady-state propane oxidation rate is governed by the extent to which its constituent steps occur simultaneously. Maximum oxidative turnover occurs within a potential window where adsorption, conversion, and oxidation are all simultaneously active. Yet, each step reaches its optimal rate at a different potential. Notably, adsorption and adsorbate conversion are maximized at mutually exclusive potentials, limiting their coexistence under constant voltage conditions. This misalignment explains the sluggish nature of steady-state alkane oxidation on Pt and highlights the need to improve the coexistence of the reaction steps to achieve higher rates.

As a proof of concept, we leveraged this insight to demonstrate that controlled potential oscillation can enhance propane oxidation by temporally promoting the conditions optimal for each principal step. More broadly, our approach enables kinetic analysis of individual steps in electrocatalytic reactions and offers a foundation for the rational design and optimization of complex electrocatalytic pathways.

## Results and discussion

### Cyclic voltammetry study of propane oxidation

We began by investigating the cyclic voltammetric (CV) oxidative stripping of propane on platinized platinum. All experiments were conducted in a stagnant thin-layer EC-MS cell in 1 M $HClO_4$ at 60 °C. The catalyst exhibited a rough, polycrystalline structure and was characterized by scanning electron microscopy (SEM), X-ray diffraction (XRD), and X-ray photoelectron spectroscopy (XPS) (Supplementary Figs. 31–34). All electrode potentials are reported relative to the standard hydrogen electrode (SHE) and are referred to as 'potentials' throughout.

Before each experiment, the electrode was cleaned by applying 1.4 V, and then 0.05 V, each for 20 s, repeated three times (Supplementary Fig. 1). The electrode was subsequently held at 0.05 V to inhibit propane adsorption and allow for MS baseline stabilization. Propane adsorption was then initiated by applying 0.3 V for durations ranging from 60 to 900 s. Previous reports have experimentally shown 0.3 V to favor adsorption while limiting adsorbate conversion[71,74,87,90–93]. Density functional theory calculations showed a 0.18 eV higher energy barrier for C−C vs C−H bond breaking in propane adsorbed on Pt(111) at 0.3 V[93], which supports the experimentally observed limited conversion of propane at this potential. Following this step, a CV was performed from the adsorption potential to 1.3 V at a scan rate of 20 mV s$^{-1}$ (Fig. 2a).

In the first cycle, three oxidative features were observed near 0.75, 0.95, and 1.25 V, denoted as Peaks I, II, and III, respectively (Fig. 2b)[71,74,94]. These features align with previous studies of alkane oxidation on Pt and are attributed to the oxidation of *CO (Peak I), partially dehydrogenated multi-carbon adsorbates (Peak II), and highly dehydrogenated or partially oxidized multi-carbon adsorbates (Peak III)[71,74,77,81,86,93,95–97]. The amplitude of each peak increased with longer adsorption times, consistent with the accumulation of *CO and multi-carbon adsorbates during the 0.3 V hold, which are subsequently oxidized to $CO_2$ at more positive potentials[90,91].

To verify that these features arise from propane-derived species, we repeated the experiments in He-saturated electrolyte. Even after 900 s of adsorption, oxidative currents were negligible under these conditions compared to those in propane-saturated electrolyte (Fig. 2b), confirming that electrolyte impurities do not significantly contribute to the observed signal. In the following sections, we build on these mechanistic insights and use EC-MS to deconvolute the rates

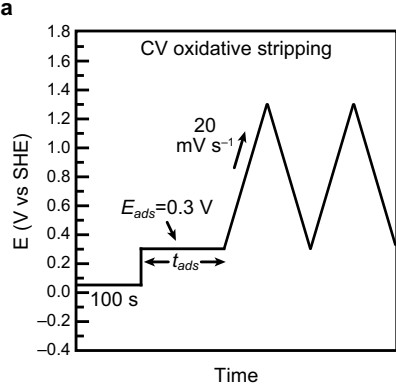

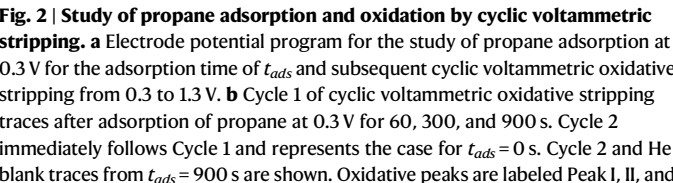

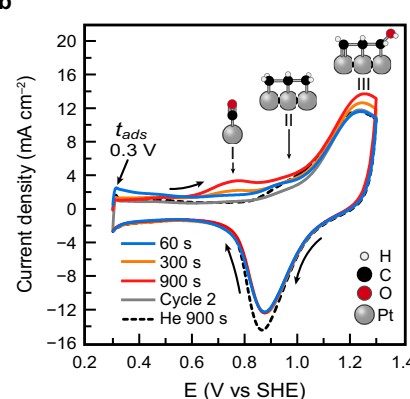

**Fig. 2 | Study of propane adsorption and oxidation by cyclic voltammetric stripping. a** Electrode potential program for the study of propane adsorption at 0.3 V for the adsorption time of $t_{ads}$ and subsequent cyclic voltammetric oxidative stripping from 0.3 to 1.3 V. **b** Cycle 1 of cyclic voltammetric oxidative stripping traces after adsorption of propane at 0.3 V for 60, 300, and 900 s. Cycle 2 immediately follows Cycle 1 and represents the case for $t_{ads} = 0$ s. Cycle 2 and He blank traces from $t_{ads} = 900$ s are shown. Oxidative peaks are labeled Peak I, II, and III where each Peak is assumed to correspond to the oxidation of the indicated adsorbate, with the intermediate shown for Peak III being hypothetical. Only one of several possible multi-carbon adsorbates corresponding to Peaks II and III are shown for simplicity. Scan rate = 20 mV s⁻¹. Current density calculated from the geometric electrode area (0.196 cm². Each CV is from a single measurement and was not iR corrected. Cell resistance = 14.90 ± 0.03 Ω. Source Data are provided as a Source Data file.

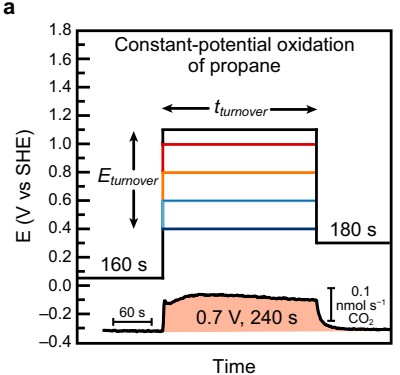

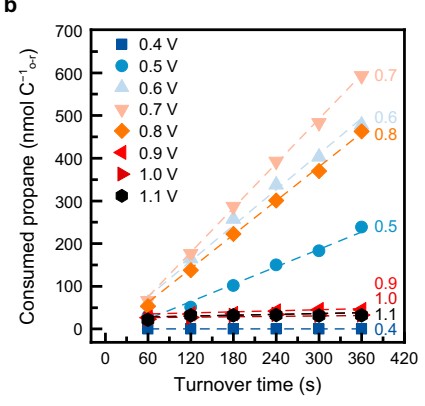

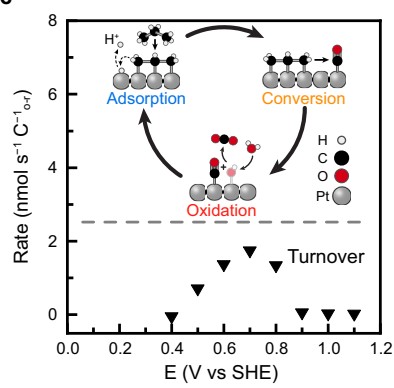

**Fig. 3 | Measurement of constant-potential propane oxidation rates.
a** Electrode potential program used to study propane constant-potential oxidative turnover. Bottom inset: $CO_2$ flux generated at 0.7 V where integrated area is shaded in orange. **b** Cumulative propane consumption as a function of potential and time. $E_{turnover} = 0.4, 0.5, 0.6, 0.7, 0.8, 0.9, 1.0,$ and 1.1 V was applied for $t_{turnover} = 360$ s. Linear fits are shown as dashed lines. Data points for a given potential are from a single measurement. **c** Rate of propane oxidative turnover as a function of the electrode potential. Top inset: illustration of the reaction steps that constitute the propane oxidative turnover cycle. Cell resistance = 100.8 ± 0.4 Ω. Source Data are provided as a Source Data file.

of the individual steps in propane oxidation, relating them to the overall steady-state oxidation rate under constant potential.

## Constant-potential oxidation of propane

The key objective of this study was to relate the overall propane oxidation rate to the rates of its individual reaction steps. To this end, the steady-state constant-potential oxidation of propane (here called "turnover") serves as the baseline for comparison. Turnover rates were measured using the procedure shown in Fig. 3a. In this procedure, EC-MS was used to quantify $CO_2$ production during the application of a constant potential '$E_{turnover}$', ranging from 0.4 to 1.1 V for 360 s in propane-saturated electrolyte. Oxidation was then halted by stepping the potential to 0.3 V and the $CO_2$ signal was monitored until it decayed to baseline. $CO_2$ evolution was monitored via the EC-MS m/z 16 signal[98–100]. Representative $CO_2$ flux traces are shown in Fig. 3a and Supplementary Fig. 2.

From these data, propane consumption was calculated using the stoichiometry of propane total oxidation (Eq. 4) and plotted as a function of time and potential (Fig. 3b). Linear fits to these data yielded

the constant-potential oxidation rates (Fig. 3c). $CO_2$ was the only observed product (Supplementary Fig. 21) and control experiments under He (Supplementary Fig. 18) confirmed that the observed $CO_2$ originated from propane oxidation rather than from electrolyte impurities[31,35,92].

In agreement with previous reports, sustained propane oxidation (i.e. "turnover") was observed between 0.5 and 0.8 V, with a maximum rate at 0.7 V[37]. This confirms that propane adsorption, conversion, and oxidation occur concurrently within this potential window. However, these data alone do not reveal which step limits the overall turnover rate at a given potential, nor do they explain why 0.7 V yields the highest activity. To address this, we next determined the rate of each individual reaction step as a function of potential, beginning with the rate of propane adsorption.

## Rate of propane adsorption

To determine the potential-dependent rate of propane adsorption, we used EC-MS to measure the amount of $CO_2$ produced when all

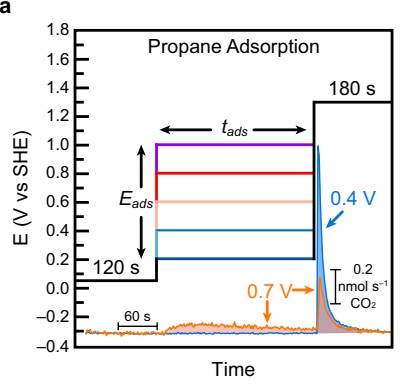
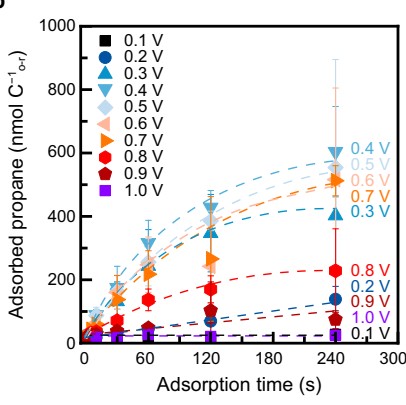
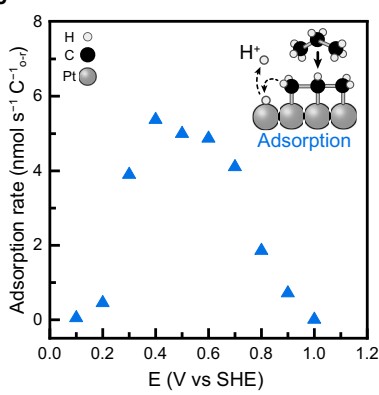

**Fig. 4 | Measurement of propane adsorption rates. a** Electrode potential program used to study propane adsorption. Bottom inset: $CO_2$ flux generated during $t_{ads} = 240$ s experiments with $E_{ads} = 0.4$ V (blue) and 0.7 V (orange), where integrated area is shaded. **b** Quantity of adsorbed propane as a function of adsorption potential and time. $E_{ads} = 0.1, 0.2, 0.3, 0.4, 0.5, 0.6, 0.7, 0.8, 0.9,$ and 1.0 V. $t_{ads} = 1, 5, 10, 30, 60, 120,$ and 240 s. Each data point between $t_{ads} = 10$ to 120 s is the average of three measurements collected on three separately prepared electrodes and

error bars show one standard deviation. Traces connecting data points were added as guides to the eye. **c** Rate of propane adsorption as a function of electrode potential calculated using $t_{ads} = 30$ s. Inset: illustration of propane adsorption. Cell resistance was adjusted to either $100.3 \pm 0.4$ Ω, $15.3 \pm 3.8$ Ω, or $12.1 \pm 0.1$ Ω to maximize potentiostat stability during EC-MS measurements. Source Data are provided as a Source Data file.

adsorbed propane was oxidized. In the experimental procedure (Fig. 4a), a potential '$E_{ads}$' between 0.1 and 1.0 V was applied to the Pt electrode for an adsorption time '$t_{ads}$' ranging from 1 to 240 s, while the electrode was immersed in propane-saturated electrolyte. After this step, the potential was changed to 1.3 V to rapidly oxidize all accumulated propane-derived intermediates and prevent further adsorption.

Representative $CO_2$ flux traces are shown in Fig. 4a and Supplementary Fig. 3. As expected from the turnover data, $CO_2$ production was also observed during the adsorption phase for $E_{ads}$ between 0.5 to 0.9 V. We contend that this $CO_2$ originates from propane that had undergone adsorption and therefore should be included in the total adsorption rate. This represents a difference from previous alkane adsorption studies, which typically quantified only the alkanes remaining after the adsorption step. While that approach helped identify adsorbates, it likely underestimates total adsorption due to omission of reactive loss during adsorption[71,74,77,88,89,93]. Reactive loss was not observed for $E_{ads}$ below 0.5 V and purging with He after propane adsorption yielded $CO_2$ quantities consistent with the ones obtained under continuous propane flow (Supplementary Fig. 20).

By integrating the $CO_2$ flux and applying the stoichiometry of propane total oxidation (Eq. 4), we calculated the amount of propane adsorbed at each potential[93]. These data are presented as a function of $t_{ads}$ and $E_{ads}$ in Fig. 4b. Control experiments with $t_{ads} = 0$ s were subtracted from the data but contributed less than 6% of the total signal measured at 0.4 V for 240 s (Supplementary Table 9) and additional controls in He-saturated electrolyte confirmed that electrolyte impurities did not significantly influence the measurements (Supplementary Fig. 17).

Adsorption rates were estimated by dividing the total adsorbed propane by $t_{ads}$ (Supplementary Fig. 4) and rates obtained at $t_{ads} = 30$ s are shown in Fig. 4c. The rate was negligible below 0.2 V, peaked at 0.4 V, decreased modestly between 0.5 and 0.7 V, and dropped sharply above 0.8 V. At longer adsorption times, adsorbate quantities began to plateau, indicating surface saturation. All measured rates were below the diffusion-limited regime (see SI), consistent with a kinetically limited adsorption process, as previously proposed by Cairns and coworkers[77]. Our measured trends differ quantitatively from earlier reports, which typically placed maximum adsorption near 0.3 V with a sharp decrease at more reductive and oxidative potentials[71,74,76,77,81,88–91,93]. We attribute these differences to

our inclusion of $CO_2$ generated during the adsorption step, made possible by real-time EC-MS measurements.

The observed potential dependence of propane adsorption and saturation can, for example, be interpreted using the water displacement theory of electrochemical adsorption, originally proposed by Bockris and collaborators[101]. According to this model, the applied potential influences neutral molecule adsorption by modulating the strength of water–electrode interactions[102,103]. Since propane must displace interfacial water during adsorption, the process is favored near the potential of zero charge (PZC), where water-electrode interactions are minimized[104]. For polycrystalline Pt, the PZC is estimated to lie near 0.3–0.4 V, consistent with the observed peak in adsorption rate[105,106]. At more reductive potentials, competitive hydrogen adsorption via underpotential deposition may inhibit propane adsorption[77,90,93].

At potentials between 0.5 and 0.8 V, the onset of dissociative water adsorption promotes the formation of *OH which has been proposed to have competing effects on propane adsorption[107–110]. Surface-bound OH (and Pt=O above 0.8 V) can block active sites, hindering further adsorption[111–113]. On the other hand, *OH can react with *CO, the intermediate formed from propane, enabling continuous oxidation[114]. This dynamic reduces adsorbate buildup and facilitates sustained adsorption. The real-time $CO_2$ detection by EC-MS allowed us to capture this effect quantitatively and incorporate it into our adsorption rates. Finally, we note that anion specific adsorption is not expected to impact our results as the perchlorate anion is believed to interact weakly with electrode surfaces[115–120].

## Rate of multi-carbon adsorbate conversion

Having established the potential-dependent rate of propane adsorption, we next examined the potential-induced conversion of adsorbed intermediates. Previous studies have shown that surface-bound alkane species undergo C−C bond cleavage and partial oxidation to form *CO[71–77]. In this work, we define the "conversion rate" as the rate at which *CO is produced from these adsorbates. Below 0.5 V, *CO accumulates on the surface, but above 0.5 V it is rapidly oxidized to $CO_2$ (see below). As such, we used different methodologies to measure conversion depending on the applied potential.

For potentials above 0.5 V, where *CO is immediately oxidized to $CO_2$, the potential program shown in Fig. 5a was used in a propane-saturated electrolyte. After pre-conditioning (Supplementary Fig. 1),

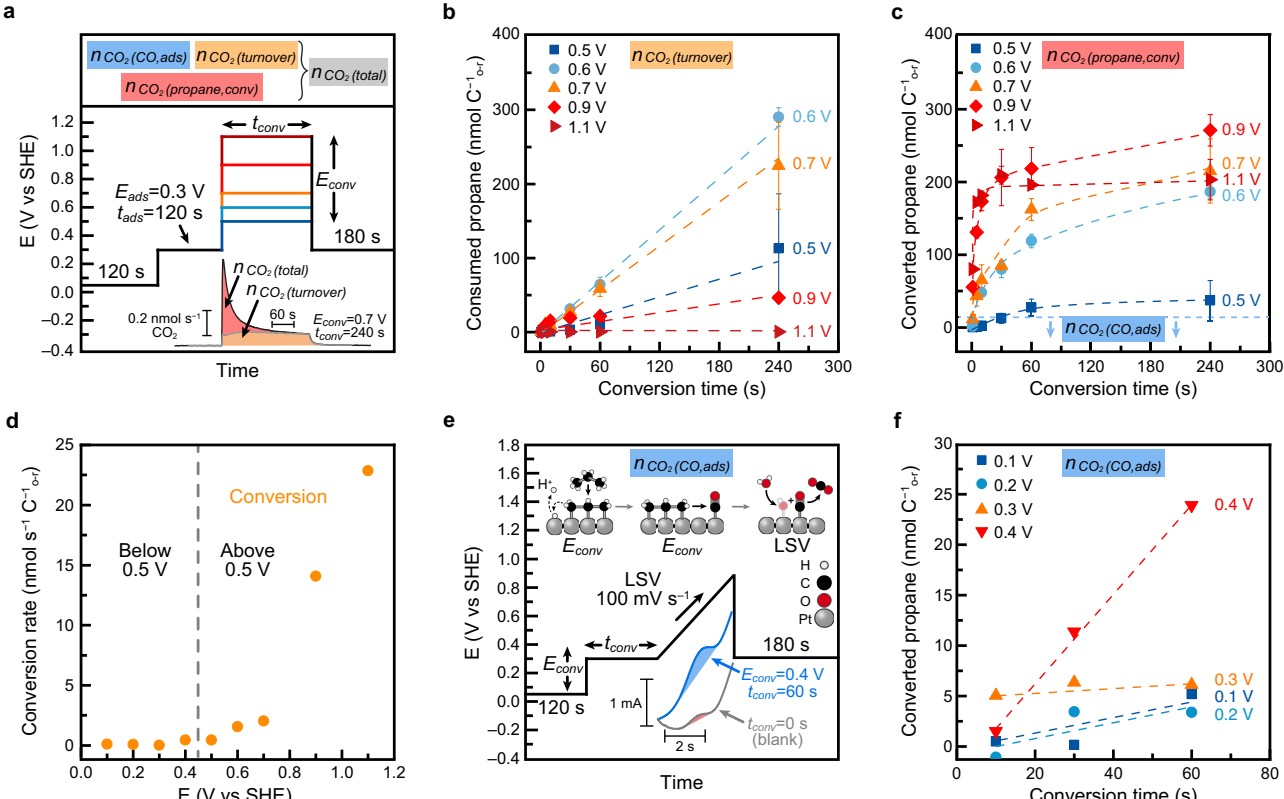

**Fig. 5 | Measurement of propane conversion rates. a** Electrode potential program used to study the conversion and oxidation of pre-adsorbed propane adsorbates for $E_{conv}$ = 0.5, 0.6, 0.7, 0.9, and 1.1 V and $t_{conv}$ = 1, 5, 10, 30, 60, and 240 s. Top inset: sources that contribute to $CO_2$ yield. Bottom inset: $CO_2$ fluxes for $E_{conv}$ = 0.7 V and $t_{conv}$ = 240 s (integrated area shaded in red) and $E_{turnover}$ = 0.7 V and $t_{turnover}$ = 240 s (integrated area shaded in orange). **b** Quantity of propane consumed due to turnover during conversion experiments. Each data point is the average of at least two measurements collected on separately prepared electrodes and error bars show one standard deviation. Dashed lines were added as guides to the eye. **c** Quantity of converted propane (turnover subtracted) as a function of $E_{conv}$ and $t_{conv}$. Each data point is the average of at least two measurements collected on separately prepared electrodes and error bars show one standard deviation. Dashed lines were added as guides to the eye. The blue dotted line at 13.3 nmol $C^{-1}_{o\text{-}r}$ represents the average quantity of propane converted to *CO during the

adsorption step. **d** Rate of propane conversion as a function of the applied potential. The gray dashed line separates the data collected using the two different methods. **e** Electrode potential program for the study of propane conversion to *CO for $E_{conv}$ = 0.1, 0.2, 0.3, 0.4 V and $t_{conv}$ = 10, 30, and 60 s. Illustrations show that adsorption and conversion occur during the application of $E_{conv}$ and *CO oxidation occurs during linear sweep voltammetry (LSV). Bottom inset: Example LSV peaks where integrated area is shaded in red for a blank and in blue for $E_{conv}$ = 0.4 V and $t_{conv}$ = 60 s experiments. Voltammetric data was not iR corrected. **f** Quantity of converted propane as a function of $E_{conv}$ and $t_{conv}$. Linear fits of the data are shown as dashed lines. Each data point is from a single measurement except for $E_{conv}$ = 0.3 V, $t_{conv}$ = 10 s where the average of two data points is reported with a standard deviation of ±0.16 nmol $C^{-1}_{o\text{-}r}$. Cell resistance = 13.9 ± 0.8 Ω. Source Data are provided as a Source Data file.

the electrode was held at 0.3 V for 120 s to adsorb propane. The potential was then stepped to '$E_{conv}$' (0.5–1.1 V) for a time '$t_{conv}$', initiating adsorbate conversion and immediate oxidation. Finally, the potential was returned to 0.3 V to terminate the reaction. The resulting $CO_2$ evolution was monitored by EC-MS ($m/z$ 16) and a representative $CO_2$ flux trace is shown in Fig. 5a. Control experiments under He (Supplementary Fig. 19) confirmed that the observed $CO_2$ originated from propane oxidation rather than from electrolyte impurities.

The total $CO_2$ produced during $E_{conv}$ includes contributions from (i) propane turnover, (ii) oxidation of *CO formed during the initial 0.3 V adsorption step, and (iii) conversion and oxidation of pre-adsorbed propane. To isolate the $CO_2$ corresponding to the conversion and oxidation of adsorbed propane '$n_{CO_2}$ (propane,conv)', we applied the following relationship:

$$n_{CO_2}(\text{propane, conv}) = n_{CO_2}(\text{total}) - (n_{CO_2}(\text{turnover}) + n_{CO_2}(\text{CO, ads})) \quad (1)$$

To determine $n_{CO_2}$ (turnover), we repeated the experiments with $t_{ads}$ = 0 s, ensuring that no pre-adsorbed intermediates were present.

The resulting $CO_2$ fluxes represent the background signal from steady-state propane oxidation at each $E_{conv}$ (Fig. 5b). These values were subtracted from the total $CO_2$ yield.

To determine $n_{CO_2}$ (CO,ads), we used linear sweep voltammetry (LSV) to oxidatively strip *CO generated during the 0.3 V adsorption step (Supplementary Fig. 6a), following methods established by Cairns and coworkers[77]. The LSV peak integrated along the baseline was converted to *CO quantity by assuming a 2 e⁻ oxidation per *CO molecule[74,86,121–124]. We expect that during LSV, further conversion and $CO_2$ generation will take place, leading to an overestimation of the conversion that occurred at 0.3 V[74,87]. Because the *CO oxidation peak in LSV is clearly distinguishable from the baseline, we chose to use the voltammetric current instead of the EC-MS signal for this analysis. Voltammograms for $t_{ads}$ = 0 s were used as blanks and subtracted from the $t_{ads}$ = 120 s signal.

With both corrections applied, we plotted the amount of converted propane $n_{CO_2}$ (propane, conv) in Fig. 5c and extracted the conversion rates (Fig. 5d) from fits to the initial linear segments of the data (Supplementary Fig. 5c). Between 0.5 and 0.7 V, conversion rates were comparable to the steady-state turnover rate but increased significantly at 0.9 and 1.1 V.

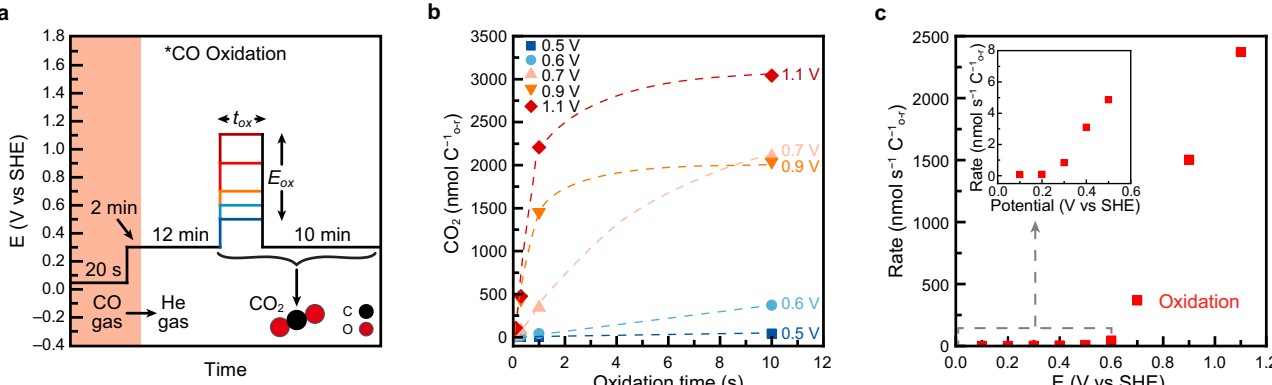

**Fig. 6 | Measurement of CO oxidation rates. a** Electrode potential program used to study *CO oxidation. **b** $CO_2$ yield as a function of oxidation potential and time. $E_{ox}$ = 0.5, 0.7, 0.9, and 1.1 V. $t_{ox}$ = 0.1, 0.3, 1, and 10 s. Traces connecting data points were added as guides to the eye. Each data point is from a single measurement. **c** Rate of CO oxidation as a function of electrode potentials. Rates below 0.5 V are from the constant-potential oxidation of solution phase CO (Supplementary Figs. 7a, b) and rates at and above 0.5 V are from *CO oxidation (Figs. 6a, b). Inset: magnification of CO oxidation rate between 0 and 0.6 V. Cell resistance = 14.5 ± 0.7 Ω. Source Data are provided as a Source Data file.

Figure 5c showed plateau-like behavior for 0.9 and 1.1 V. We interpret this as the rapid conversion and oxidation of most of the pre-adsorbed propane. However, the amount of propane converted during $t_{conv}$ fell short of the total adsorbed quantity, even at the most oxidative potentials. This suggests that not all pre-adsorbed propane was converted during these experiments. Additional experiments, in which 1.3 V was applied immediately after $E_{conv}$, confirmed the presence of unreacted adsorbates (Supplementary Fig. 12). This observation aligns with previous reports of recalcitrant intermediates that persist on the surface and are unlikely to contribute to propane oxidative turnover[74,77]. Importantly, the corrected yields remained within error of the expected pre-adsorbed amount, supporting the validity of our turnover correction approach.

For $E_{conv}$ < 0.5 V, *CO accumulates instead of being oxidized, allowing direct quantification by LSV oxidative stripping. Using the procedure shown in Fig. 5e, we determined *CO yields for various values of $E_{conv}$ and $t_{conv}$. As a validation step, we compared blank ($t_{ads}$ = 0 s) LSV traces to the shortest conversion time experiments and found similar signals (Supplementary Fig. 11), confirming minimal conversion at early times. An example LSV at 0.4 V and 60 s is shown in Fig. 5e. Converted propane amounts, as derived from integrated *CO quantities, (corrected for the blank) are shown in Fig. 5f, and conversion rates extracted from linear fits are included in Fig. 5d. Below 0.5 V, conversion was slow but accelerated slightly from 0.1 to 0.4 V. Having quantified the potential-dependent rate of *CO formation, we next investigated the kinetics of *CO oxidation to determine how this final step regulates overall propane turnover.

## Rate of CO oxidation
To determine the rate of *CO oxidation, we used CO as the substrate and studied its oxidation as a function of the electrode potential. For potentials between 0.1 and 0.4 V, oxidation rates were obtained from constant-potential experiments in CO-saturated electrolyte over 360 s (Supplementary Fig. 7). At higher potentials, mass transport limitations required an alternative approach based on pre-adsorbed *CO. In this method (Fig. 6a), a partial *CO monolayer was generated by exposing the cleaned electrode to a CO-saturated electrolyte at 0.3 V for 120 s. The electrolyte was then purged with He to remove dissolved CO, and *CO was oxidized by stepping the potential to values between 0.5 and 1.1 V for durations between 0.1 and 10 s. The resulting $CO_2$ yield was plotted as a function of time and potential (Fig. 6b) and oxidation rates were extracted from linear fits to the initial yields between 0.1 and 1 s (Supplementary Fig. 8). Combined results from both methods are shown in Fig. 6c.

CO oxidation rates increased gradually between 0.2 to 0.6 V, then rose more sharply at more oxidative potentials. The plateauing behavior of the $CO_2$ yields in Fig. 6b indicates that *CO was rapidly consumed under oxidative conditions[123,125]. Notably, we detected non-zero $CO_2$ production from CO oxidation at 0.3 and 0.4 V. While propane oxidative turnover was not observed in this potential range, the shared *CO intermediate suggests that steady-state propane turnover may nonetheless be possible at these lower potentials. However, given the low *CO formation rates from propane observed at 0.3–0.4 V, *CO likely only accounts for a small fraction of surface adsorbates, which may explain why $CO_2$ production from propane remains below the EC-MS detection limit under these conditions. Taken together with the results of the previous sections, this analysis completes the mechanistic decomposition of propane oxidation and enables a step-resolved comparison of potential-dependent rates across the full reaction sequence.

## Identification of propane oxidation bottlenecks
The potential-dependent rates of each step in propane oxidation allow us to identify the rate-limiting steps across the potential range studied. Independent measurements of the turnover, adsorption, conversion, and oxidation rates (Figs. 3c, 4c, 5d, and 6c) were compared in Fig. 7a. At each potential, the slowest step can be interpreted as rate-limiting, while the degree of overlap among the rates of the principal reaction steps provides insight into the overall turnover behavior. This interpretation is supported by the close match between the rate of the slowest step and the measured turnover rate across the potential range. Multiple potential windows were identified in which different reaction steps limit the overall reaction rate, as summarized in Fig. 7b. Below, we describe the bottlenecks in order of increasing potential.

CO Oxidation: *CO oxidation is the slowest step at 0.2 V and below, where it limits the overall rate of propane oxidation. Between 0.3 and 0.5 V, the oxidation rate increases and becomes the fastest step at 0.6 V and above. The oxidation of *CO is commonly described by a Langmuir-Hinshelwood-Hougen-Watson mechanism in which *OH and *CO combine at the surface to form $CO_2$ gas[114,123,126]. The observed potential-dependence aligns with the onset of *OH formation on Pt in acidic media[107,109,110]. Our finding that CO oxidation is no longer rate-limiting above 0.2 V is consistent with the recent results of Kong and coworkers who showed that introducing CO, commonly thought of as a catalyst poison, to the fuel supply of a propane fuel cell did not reduce performance[48].

Adsorbate Conversion: The conversion of multi-carbon adsorbates to *CO is rate-limiting between 0.3 and 0.7 V, the potential range

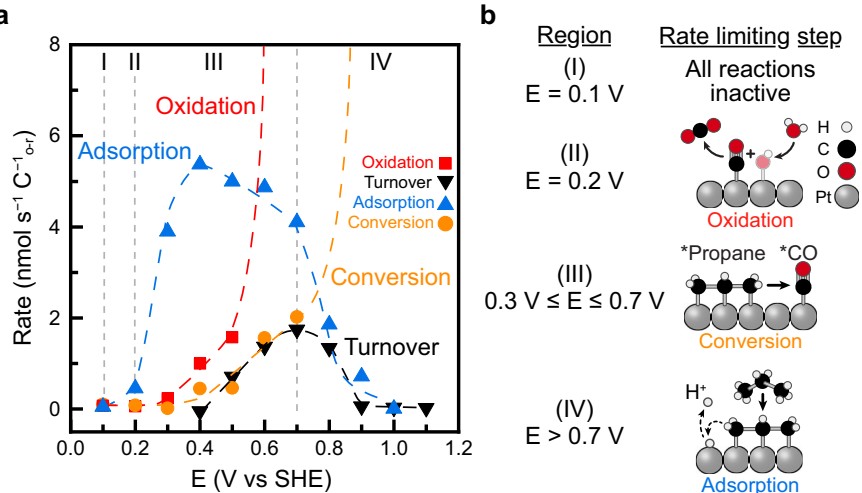

**Fig. 7 | Comparison of continuous turnover to the rate of individual steps.** **a** Rate of each principal step of propane total oxidation measured independently as a function of the electrode potential. Potential regions where different reaction steps limit the overall rate of reaction are labeled. Dashed lines were added as guides to the eye. **b** Potential ranges and rate-limiting reaction for the labeled regions. Source Data are provided as a Source Data file.

where steady-state alkane oxidation exhibits the highest overall rates. This indicates that, under typical operating conditions of a Pt-based propane fuel cell, the conversion step represents the primary kinetic bottleneck. It is important to note that rate-limiting conversion ultimately leads to multi-carbon adsorbate accumulation on the surface, which we expect to influence both adsorption and oxidation reactions[46,47,77,91].

Alkane Adsorption: Above 0.7 V, propane adsorption becomes rate limiting. We attribute this to two effects. First, at potentials more oxidative than the PZC, the Pt surface becomes increasingly positively charged, promoting the recruitment of anions and water molecules to the electrode surface and thereby reducing the availability of propane at the interface[101,104,127–133]. Second, at potentials above 0.8 V, platinum oxide begins to form[107,108]. Prior studies have shown that the energy barrier for alkane adsorption on Pt oxide is more than 0.5 eV higher than on metallic Pt[111], significantly reducing propane adsorption rates[112,113].

These results demonstrate that the conditions favoring each step in propane oxidation are often mutually exclusive. On Pt electrodes, the potentials that promote fast conversion and *CO oxidation suppress propane adsorption. Conversely, conditions favorable for substrate adsorption do not allow for high rates of conversion and oxidation. The maximum turnover rate at 0.7 V reflects a compromise, where the partial overlap of all three steps enables optimal performance.

This mechanistic insight defines key design criteria for improving alkane oxidation: a good catalyst should enable rapid adsorbate conversion and *CO oxidation within the potential range where adsorption is also favorable. In the following section, we demonstrate how applying alternating electrode potentials can help reconcile these competing requirements and enhance overall propane oxidation.

### Potential pulses for the oxidation of propane

Achieving high rates of propane oxidation requires a reaction environment optimized for each catalytic step. Our analysis revealed that the potential at which adsorption is maximized differs from the potentials that promote rapid adsorbate conversion and *CO oxidation. Additionally, adsorption was found to be fastest at short times, slowing as surface coverage increased. These observations suggest that carrying out each step at its optimum potential, rather than under a single steady-state condition, could enhance the overall rates.

As shown in Supplementary Fig. 4 the maximum rate of propane adsorption occurs at 0.4 V, reaching 7.4 nmol $s^{-1}$ $C^{-1}_{o-r}$. In contrast, both conversion and oxidation rates increase monotonically with potential. To reconcile the differing optimal conditions for each step, we designed an alternating potential program that enables adsorption at low potential and conversion/oxidation at high potential.

The applied pulse sequence is illustrated in Fig. 8a. After electrode pre-conditioning (Supplementary Fig. 1) and baseline stabilization at 0.05 V, the potential was alternated between an adsorption potential '$E_{ads}$' and an oxidation potential '$E_{ox}$' for durations '$t_{ads}$' and '$t_{ox}$', respectively. After 180 cycles, $CO_2$ production was halted by lowering the potential to 0.3 V. The total amount of $CO_2$ produced was used to calculate propane consumption during pulsed operation (Fig. 8b), which was then compared to constant-potential oxidation at 0.7 V.

Under steady-state conditions at 0.7 V, the propane oxidation rate was 1.7 nmol $s^{-1}$ $C^{-1}_{o-r}$. In contrast, the alternating potential protocol with $E_{ads} = 0.4$ V, $E_{ox} = 0.9$ V, and $t_{ads} = t_{ox} = 1$ s yielded an oxidation rate of 2.9 nmol $s^{-1}$ $C^{-1}_{o-r}$. This result demonstrates that the kinetic bottlenecks of constant-potential operation can be mitigated through rational application of time-dependent potentials that separately optimize the conditions for each reaction step. Similar strategies have been successfully employed in formic acid oxidation, as demonstrated by Adžić, as well as Gopeesingh[134,135].

The ability to identify and decouple the contributions of adsorption, conversion, and oxidation was essential to designing this enhancement strategy. Building on this approach, we are now exploring methods to further improve alkane adsorption rates and to tune the potential windows for conversion and oxidation steps in electrocatalytic hydrocarbon reactions.

This study provides a comprehensive analysis of the steady-state oxidation of propane to $CO_2$, showing that the overall rate is governed by the degree of overlap between the rates of its principal steps: adsorption, conversion, and oxidation. Using electrochemical mass spectrometry, we quantified the rate of each of these steps independently and directly compared them to the steady-state turnover rate. Our findings reveal that low propane oxidation rates arise from the mutual exclusivity of the potential ranges that promote fast adsorption versus those that favor rapid conversion and oxidation. As a proof of concept, we applied alternating potentials to individually optimize each step, achieving propane oxidation rates exceeding those attainable under constant-potential conditions.

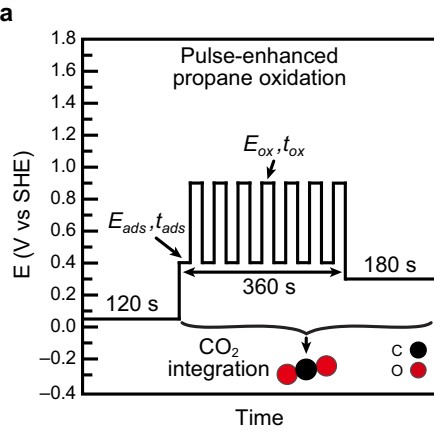

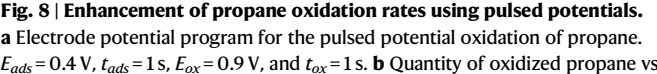

**Fig. 8 | Enhancement of propane oxidation rates using pulsed potentials.** **a** Electrode potential program for the pulsed potential oxidation of propane. $E_{ads} = 0.4\,V$, $t_{ads} = 1\,s$, $E_{ox} = 0.9\,V$, and $t_{ox} = 1\,s$. **b** Quantity of oxidized propane vs experiment time. Linear fits of the data are shown as dashed lines. Data points for a given condition are from a single measurement. Cell resistance = $100.9 \pm 0.5\,\Omega$. Source Data are provided as a Source Data file.

These results highlight the critical importance of step-resolved mechanistic insight in the rational design of electrocatalytic systems. By understanding and addressing kinetic bottlenecks, we enable targeted strategies to improve performance. This framework opens new pathways for the development of advanced electrocatalysts for fuel cells, energy conversion, and chemical technologies, advancing progress toward a future energy system.

## Methods

### General procedure

Except where noted, all experiments were carried out using platinized Pt catalysts and 1 M perchloric acid electrolyte at $60 \pm 1\,°C$. Unless otherwise stated, all experiments were performed in a propane-saturated electrolyte with a propane flow rate of 1 sccm. $CO_2$ yields were quantified from the $m/z$ 16 MS signals. Voltammetry and electrochemical mass spectrometry experiments were performed using the electrochemical protocols described in the text. Each experiment was preceded by an electrode cleaning step at 1.4 V, followed by 0.05 V for 20 s each, repeated 3 times (Supplementary Fig. 1). Additional details are provided in the Supplementary Information.

### Materials: gases

Helium gas (Ultra High Purity Grade), Propane (Research Grade), $H_2$ (Ultra High Purity 5.0 Grade), $CO_2$ (Research Grade), and CO (Research Plus Grade) were purchased from Airgas.

### Cleaning

All glassware and polytetrafluoroethylene (PTFE) electrochemical cell components used in the preparation of electrolytes and during experiments were cleaned using piranha solution formulated from 85% $H_2SO_4$ (Sigma-Aldrich, ACS Reagent grade, 95.0-98.0%) and 15% $H_2O_2$ (Honeywell, semiconductor grade, 30% or Fisher, ACS grade, 30%) then rinsed thoroughly with DI water (ACS Reagent Grade, ASTM Type 1, LabChem). Glass electrochemical cell components were dried in an oven at 80 °C and then cooled to room temperature under ambient conditions before use. PTFE electrochemical cell components were dried with compressed air before use.

### Electrolyte

Perchloric acid (70%, 99.999% trace metals basis) was purchased from Millipore-Sigma. 1 M perchloric acid was used as the electrolyte for all reported experiments. This solution was prepared by diluting 70% perchloric acid with purified water (18.2 MΩ, Millipore, Milli-Q

reference A+, TOC = 2 ppb). Adjustable micropipettes (Eppendorf, Research Plus) and volumetric glassware were used to prepare the diluted perchloric acid. 500 mL of solution was prepared and stored in a 500 mL glass bottle (piranha solution cleaned) with polypropylene cap (VWR). Before use, approximately 10 mL of 1 M perchloric acid was transferred to a 50 mL sterile centrifuge tube (Falcon, Corning). This solution was then sparged with He for at least 20 min using a glass gas-dispersion tube (piranha solution cleaned) to remove dissolved gas. Three batches of electrolyte were used during this work and were all prepared and stored in the same way. The first batch was prepared on 10/29/2023 and had a pH value of $0.15 \pm 0.01$, the second batch was prepared on 1/3/2024 and had a pH value of $0.16 \pm 0.02$, and the third batch was prepared on 4/2/2024 and had a pH value of $0.15 \pm 0.06$. Measurements of pH were performed using a Mettler-Toledo FiveGo F2 with LE438 probe.

### Electrodes

A platinized Pt electrode was used as the working electrode. A coiled Pt wire (Kurt J. Lesker, 99.99%), flame cleaned with a butane torch, and a Ag/AgCl electrode (BASi, RE-5B, 3.0 M KCl) were used as the counter and reference electrodes, respectively. Reference electrodes were calibrated by measuring the potential difference between an identical reference electrode which was never used in experiments and assumed to have a value of 0.210 V vs SHE.

### Platinized Pt catalyst preparation

A 5 mm diameter platinum disk electrode (99.995%, Pine Research) was polished using progressively finer alumina slurries (1.0, 0.3 and 0.05 μm, Allied High Tech) and sonicated (Braunsen 3800) in DI water (ACS Reagent Grade, ASTM Type 1, LabChem) after each polishing step. High surface area platinized platinum was then deposited via electro-deposition from a solution of 0.072 M $H_2PtCl_6$ (99.9% trace metal basis, Millipore-Sigma) and $1.3 \times 10^{-4}$ M $Pb(C_2H_3O_2)_2$ (99.999% trace metal basis, Millipore-Sigma) by applying $-30\,mA\,cm^{-2}$ ($-5.89\,mA \div 0.196\,cm^2_{geo}$) for 500 s[74,136]. The application of current was controlled by a Biologic SP-200 potentiostat with EC-lab (Version 11.33) software. Catalyst deposition was performed in a two-electrode configuration with a Pt wire (Kurt J. Lesker, 99.99%), flame cleaned with a butane torch, used as the counter electrode. Electrodeposition was performed at room temperature. After deposition, the catalyst was submerged in DI water (ACS Reagent Grade, ASTM Type 1, LabChem) several times to rinse off residual platinizing solution. The catalyst loading was approximated to be $2.19\,mg\,cm^{-2}$ by recording the mass (Mettler-

Toledo, XSR205) difference of the 0.196 $cm^2_{geo}$ surface area electrode before and after platinizing, rinsing, and drying the electrode. Small particles of deposited Pt were often dislodged during the rinsing step which affects catalyst loading for each electrode preparation. To account for this and for changes in the electroactive surface area during successive electrode use, all voltammetric current and ionic current signals were normalized by dividing the signals by the absolute value of the charge required to reduce the platinum oxide ($C_{o-r}$) formed during the second CV cycle performed from 0 to 1.4 V at 50 mV s$^{-1}$ after experiment completion[137]. An example CV with charge integration is shown in Supplementary Fig. 14. Representative electrochemical surface area (ECSA) measurements were performed and approximated to be $108 \pm 4$ cm$^2$ by analyzing the oxidative charge associated with the desorption of hydrogen in the hydrogen underpotential deposition ($H_{UPD}$) region calculated using a factor of 210 μC cm$^{-2}$ (Supplementary Fig. 35a)[95]. This ECSA value agrees with $100 \pm 8$ cm$^2$ estimated by analyzing the He background subtracted charge passed during the oxidation of a monolayer of CO adsorbed to the electrode at 0.1 V for 20 min and calculated using a factor of 420 μC cm$^{-2}$ (Supplementary Fig. 35b)[95]. These ECSA measurements are associated with a $C_{o-r}$ value of $-0.0278 \pm 0.001$ C (Supplementary Fig. 35a).

Each electrode was pre-conditioned using 10 cycles of cyclic voltammetry (CV) at 50 mV s$^{-1}$ from 0 to 1.4 V vs SHE in He-saturated 1 M perchloric acid at room temperature. An additional 10 CV cycles in a He-saturated electrolyte at the reaction temperature and 20 cycles in a substrate-saturated electrolyte at the reaction temperature were performed before performing experiments. A typical CV for the EC-MS set-up is shown in Supplementary Fig. 9. The Supplementary Information includes catalyst characterization using X-ray diffraction (XRD), X-ray photoelectron spectroscopy (XPS), and scanning electron microscopy (SEM).

## Electrochemical mass spectrometer setup
A Spectro Inlets (Denmark) Electrochemical Mass Spectrometer controlled by Zilien software (Spectro Inlets, Denmark, Version 2.5.0) was used to perform all EC-MS measurements[4]. All EC-MS experiments were conducted in a PTFE stagnant thin-layer electrochemical cell which was interfaced with the membrane chip MS inlet (aqueous chip, Spectro Inlets) as shown in Supplementary Figs. 36, 37, 38, and 39. Electrolyte was injected into the electrochemical cell using single-use Henke-Ject syringes (HENKE SASS WOLF, Luer Lock, sterile, 3 mL, 4020-X00V0). The approximate volumes of electrolyte in the working, counter, and reference electrode compartments were 9.5 μL, 1 mL, and 2 mL, respectively. The cell compartments were not separated by frits or membranes. The electrochemical cell and cell mounting block were heated to $60 \pm 1$ °C for all experiments unless otherwise stated using heating tape (BriskHeat), controlled by a BriskHeat controller (SDXJA), monitored using a thermocouple (HH802U, OMEGA Engineering) positioned between the electrochemical cell and the EC-MS cell mounting block, and insulated with glass wool.

When using propane as the substrate, $CO_2$ signals were quantified from the m/z 16 signal due to large overlap between $CO_2$ and propane m/z 44 MS signals. MS signals were not processed using a deconvolution protocol. MS ionic current signals and peaks were integrated in reference to the signal baseline measured after experiment completion using Origin Lab (Version 2022b). All gases were supplied via the Spectro Inlets EC-MS gas manifold mass flow controllers and introduced to the electrolyte through the gas-permeable membrane chip. During experiments that involved mid-experiment gas switching (Oxidation of *CO, Fig. 6), gases were initially flown at 10 sccm to facilitate rapid removal of the previously supplied gas from the electrolyte, then flown at 1 sccm during the last 1 min of a gas exchange step. All gases were flown at 1 sccm during all other processes.

Applied potentials were controlled by a Biologic SP-200 potentiostat with EC-lab (Version 11.33) software interfaced with the Zilien

(Version 2.5.0) EC-MS software. Either a 10 Ω, 30 Ω, or 100 Ω resistor was connected in series with the working electrode to improve potentiostat signal stability. Potentiostatic electrochemical impedance spectroscopy from 200 to 200,000 Hz with an amplitude of 10 mV was performed before each EC-MS experiment. Cell resistance is reported for relevant figures and tables. Due to low faradaic currents, iR values were considered negligible and data were not iR corrected.

## Electrochemical mass spectrometer calibration
The EC-MS system was calibrated so that ionic currents could be converted to analyte flux. First, a calibration factor to calculate analyte flux from analyte concentration was determined[85]. This was accomplished by performing both internal and external calibrations for $H_2$ gas. Internal calibration of $H_2$ was completed by measuring the m/z 2 ionic current while producing $H_2$ via the hydrogen evolution reaction (HER) in 1 M perchloric acid. A two-electrode setup with a polished Pt disk (99.995%, Pine Research) and a Pt wire (99.99%, Kurt J. Lesker) were used as the working and counter electrodes, respectively. Applied currents were controlled by a Biologic SP-200 potentiostat with EC-lab (Version 11.33) software interfaced with the Zilien EC-MS software. Assuming 100% Faradaic efficiency for $H_2$ production, constant-current experiments from −1 to −10 μA were performed to generate $H_2$ at a known rate (nmol s$^{-1}$) and the corresponding m/z 2 ionic currents (A) were measured (Supplementary Fig. 13a). The employed EC-MS system is reported to have a 100% collection efficiency, therefore under steady-state $H_2$ production, the $H_2$ production rate is expected to be equivalent to the $H_2$ flux (nmol s$^{-1}$) reaching the MS for detection. Leveraging this 1:1 relationship, a calibration curve relating the m/z 2 ionic current to the $H_2$ flux was constructed (Supplementary Fig. 13b).

An external $H_2$ calibration was then performed where the m/z 2 ionic current was measured when varying concentrations of $H_2$ were introduced to the EC-MS system while the electrochemical cell was mounted and filled with Milli-Q water. This was accomplished by using $H_2$, diluted in He from 5000 to 25,000 ppm, as the carrier gas. The m/z 2 ionic current vs $H_2$ concentration data was plotted in Supplementary Fig. 13c.

Next, the data from both the internal and external $H_2$ calibrations were used to relate concentration to flux. Assuming that the flux of He is constant for dilute mixtures, the total flux of gas to the MS vacuum chamber was calculated as

$$Total\,Gas\,Flux = \frac{\left(S^{m/z2}_{x_{H_2}} - b\right)/m}{x_{H_2}}, \qquad (2)$$

where $S^{m/z2}_{x_{H_2}}$ is the m/z 2 MS ionic current signal obtained during the hydrogen external calibration for the hydrogen concentration $x_{H_2}$, $b$ is the intercept from the $H_2$ internal calibration (m/z 2 ionic current background), $m$ is the slope of the $H_2$ internal calibration, and $x_{H_2}$ is the mole fraction of $H_2$ from the external calibration. Using Eq. 2, the total flux of gas reaching the MS was calculated for each concentration of $H_2$ introduced during the external calibration. An average value of 7.62 nmol s$^{-1}$ was determined. This value was used for the conversion of concentration to flux for other externally calibrated gases.

To quantify $CO_2$ produced during EC-MS experiments, an external calibration was performed by introducing $CO_2$ between 400 and 250,000 ppm, diluted with He, to the EC-MS system. The mole fraction of $CO_2$ introduced to the system was then multiplied by 7.62 nmol s$^{-1}$ to obtain the flux of $CO_2$ (Supplementary Fig. 14a). This conversion allowed a calibration curve of m/z 16 ionic current vs $CO_2$ flux to be constructed (Supplementary Fig. 14b). Performing a linear fit of this data yielded a calibration factor of $5.91 \times 10^{-10}$ A s nmol$^{-1}$. This calibration factor was used to convert the m/z 16 ionic current measured during each experiment to the flux of $CO_2$.

During calibration, the $m/z$ 4 ionic current while delivering 1 sccm He carrier gas to the EC-MS was recorded. For every set of experiments, the $m/z$ 4 ionic current, while delivering 1 sccm He carrier gas to the EC-MS, was also recorded and referenced to the measurement recorded during calibration. A He chip correction factor was then implemented to account for slight variability in the MS tuning, secondary electron multiplier signal enhancement, atmospheric pressure, and membrane chip capillary volume. The He chip correction factor was calculated as

$$\text{He chip correction factor} = \frac{m/z\ 4\ ionic\ current\ (calibration)}{m/z\ 4\ ionic\ current\ (experiment)} \quad (3)$$

This correction factor was determined for every set of experiments. The calibration factor for $CO_2$ was multiplied by the determined He chip correction factor before using it to convert $m/z$ 16 ionic current to $CO_2$ flux. This correction was used in previous publications from our group and is similar to that performed by others[74,85–87,138,139]. All EC-MS calibration experiments were performed at room temperature.

### Constant-potential CO oxidation

To calculate the *CO oxidation rate below 0.5 V, the constant-potential oxidation of solution phase CO was measured as a function of the electrode potential using the procedure shown in Supplementary Fig. 7a. Each experiment was preceded by an electrode cleaning step at 1.4 V, followed by 0.05 V for 20 s each, repeated 3 times (Supplementary Fig. 1). After cleaning, the MS signal was allowed to stabilize for 600 s at 0.05 V. The potential was then increased to $E_{ox}$ for 360 s. Next, the potential was decreased to 0.05 V to allow the MS signal to stabilize for 600 s. The $m/z$ 16 ionic current during $E_{ox}$ and the following 0.3 V stabilization period were integrated. $CO_2$ yields were calculated from these integrals and the cumulative $CO_2$ yields during $E_{ox}$ were plotted in Supplementary Fig. 7b. The CO oxidation rate was calculated for each potential by dividing the $CO_2$ yield by the 360 s oxidation time and data are shown in Fig. 6c. The corresponding CO oxidation rates were expressed in terms of propane oxidation rates by dividing by the stoichiometric coefficient of 3, according to Eq. 4, before plotting in Fig. 7a.

### Calculation of reaction rates

In this study, $CO_2$ yields were used to measure the rate of propane surface reactions. The rates of propane consumption, adsorption, conversion, and oxidation were calculated by dividing the $CO_2$ yields by the stoichiometric coefficient of 3 according to the total oxidation of propane:

$$C_3H_8 + 6\,H_2O \rightarrow 3\,CO_2 + 20\,H^+ + 20\,e^- \quad (4)$$

## Data availability

The data plotted, interpreted, and discussed above and in the Supplementary Information is available in the Source Data file. Source data are provided with this paper.

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

## Acknowledgements

A.J.Z. acknowledges a Seed Grant through the Wisconsin Materials Research Science and Engineering Center (DMR-1720415) and funding through an NSF CAREER Award (CBET-2338627). We acknowledge use of facilities and instrumentation at the UW–Madison Wisconsin Centers for Nanoscale Technology, partially supported by the NSF through the University of Wisconsin Materials Research Science and Engineering Center (DMR-1720415). M.S. acknowledges support by the Arnold and Mabel Beckman Foundation through a Beckman Young Investigator Award (https://doi.org/10.13039/100000997, M.S.). C.L. acknowledges funding from the National Science Foundation Graduate Research Fellowship Program under Grant No. DGE- 2137424. Any opinions, findings, and conclusions or recommendations expressed in this material are those of the authors and do not necessarily reflect the views of the National Science Foundation. Support was also provided by the Graduate School and the Office of the Vice Chancellor for Research and

Graduate Education at the University of Wisconsin–Madison with funding from the Wisconsin Alumni Research Foundation. We thank Lee Fuller for performing the SEM measurements, Geunryeol Baek for performing the XPS measurements, Gong Zhang for helpful discussion, and Megan Kelly and Enner Mendoza for input.

## Author contributions

A.Z. and M.S. conceived the project. A.Z. designed and performed experiments. C.L. assisted with data collection. A.Z., C.L., and M.S. analyzed the data. A.Z. and M.S. wrote the paper with C.L. contributing to revisions. M.S. provided funding acquisition and project supervision.

## Competing interests

The authors declare no competing interests.
