## [Transparent Peer Review file · Nature Communications]

Linking Electrocatalytic Turnover to Elementary Step Rates in Hydrocarbon Fuel Oxidation

Corresponding Author: Professor Marcel Schreier

This file contains all reviewer reports in order by version, followed by all author rebuttals in order by version. Parts of this Peer Review File have been redacted as indicated to maintain the confidentiality of unpublished data.

Version 0:

Reviewer comments:

Reviewer #1

(Remarks to the Author)

The reviewers have conducted a thorough and comprehensive review of the manuscript. Based on their responses and edits, the quality and clarity of the article have been significantly improved. They have also addressed all of the reviewers comments. I support publication of this article in Nature Communications, where its interdisciplinary readership would likely provide a more appropriate audience and reception for this research.

Reviewer #2

(Remarks to the Author)

The authors have addressed my technical concerns, and this manuscript is suitable for publication Nature Comm.

Reviewer #4

(Remarks to the Author)

Comment 1.1. The authors have implemented all the comments from the previous reviewer. Thus, additional physicochemical characterization (SEM, XRD and XPS) and additional blank EC-MS experiments have been added to the manuscript. The same has also been carefully revised in terms of writing throughout the whole article. However, this reviewer understands that comment 1.4 hasn't been sufficiently well explained. Therefore, this reviewer considers the article to be published after addressing the cited comment.

Comment 1.2. The authors have added to the manuscript an extensive physicochemical characterization of the Pt-based electrode. This includes Scanning Electron Microscopy, X-ray Diffraction and X-Ray Photoelectron Spectroscopy. The diffraction patterns show the polycrystalline nature of the sample along with a crystallite size of 13 nm, which is also aligned with the SEM results shown in Figure S34(f). The XPS results show the expected signals for a Pt-based materials, besides some other contributions (chlorine and fluor) which have been assigned to traces from both the synthetic procedure and the handling and transferring of the sample. Thus, this sheds some light on the characterization of the material and the comment is considered to be solved.

Comment 1.3. The authors have revisited Figure 2(b) of the main article to show all the experiments recorded in the same conditions ($T = 60\text{ }^{\circ}\text{C}$, He-bubbling and in the same cell as the DEMS experiments shown in the article). Besides this, the previous experiments ($T = 80\text{ }^{\circ}\text{C}$) have been moved to the SI and the text has been accordingly revised. Thus, this comment is considered to be solved.

Comment 1.4. The authors now have explained the reason to adsorb the propane at 0.3 VSHE. Based on previous studies (ref 74 and 87 from the main article, both are citations from the same research group) they concluded that this potential is the best for favoring adsorption while limiting adsorbate conversion. Thus, this reviewer understands that the reason behind choosing 0.3 VSHE is based on previous experience for the same experiments with other alkanes (ethane and butane). Is this statement supported by theoretical calculations? This reviewer suggests to slightly extend or clarify if the precise value

of 0.3 VSHE exclusively rises from experimental measurements or if there are any calculations or different literature explaining this effect.

Comment 1.5. The authors have now added to the supporting information of the article the blank DEMS experiments (He-saturated) to discard activity coming from the oxidation of impurities contained in the HClO₄ electrolyte solution. The new data-sets (Figures S17, S18 and S19) now contain also the blanks where it is clearly visible that the activity is very low (authors claimed it to be under 6% in comparison with the target experiment). The authors stated that this does not influence the results, and they have also accordingly revised the text at both the main article and the supporting information. This reviewer agrees and the comment is considered to be solved.

Comment 1.6. The authors have substantially revised the manuscript to improve the clarity. In the present form, the findings of the work and the main conclusions may be more easily followed by the reader. Thus, this comment is also considered to be solved.

Response to Reviewers

Reviewer #1:

Comment 1.1: In this article, the authors investigate the electrocatalytic oxidation of propane by analyzing the rates of individual reaction steps using electrochemical mass spectrometry (EC-MS). The authors identify three key steps—adsorption, conversion to CO, and CO oxidation—and show that their optimal conditions do not overlap, limiting overall efficiency. While the highest oxidation rate occurs at 0.7 V, adsorption is fastest at lower potentials, and conversion and oxidation improve at higher potentials. To address this mismatch, they apply a dynamic pulsing strategy, alternating potentials to enhance reaction rates beyond steady-state conditions. This work provides insights for optimizing hydrocarbon fuel cells and electrocatalytic systems.

While the work presents important mechanistic insights, there are several key limitations to consider in evaluating its suitability for publication in Nature Energy.

Response 1.1: We thank the reviewer for the careful analysis of our work and the provided feedback.

Comment 1.2: While the authors presented a clever approach to deconvolute the effect of individual steps in the propane oxidation reaction mechanism, the paper's impact is likely limited since some findings have already been demonstrated previously by the authors: DOI: 10.1039/d3cy01172k showed CO is the dominant intermediate on Pt via mass spec analysis, with maximum methane adsorption at 0.3V; DOI: 10.1021/jacs.3c02108 demonstrated pulsed potentials can improve control over ethane C-C bond breaking, varying steps between adsorption (0.3V) and oxidative fragmentation; DOI: 10.1038/s41929-024-01218-0 identified butane forms butyl fragments and CO species with fragmentation kinetics limiting performance above 0.6V and CO oxidation limiting below 0.5V. What is unique about this work is that its quantification of all three steps (adsorption, conversion, oxidation) vs potential reveals that they occur at mutually exclusive potentials, and moreover that the maximum turnover at 0.7 V represents optimal overlap between steps, explaining inherent limitations of constant-potential operation.

Response 1.2: We appreciate the reviewer's positive comment on our approach. We agree with the reviewer that the independent quantification of the rate of all three reaction steps leads to unique insight, demonstrating that the individual steps of this reaction occur under mutually exclusive conditions.

Our assessment is that our work is impactful for the following reasons:

- 1) A key novelty of our work is that it leverages the pulsed-potential approach we have published on (as mentioned by the reviewer) to *deconvolute the rate of a continuously turning over reaction* (the low-temperature oxidation of propane) into the rates of its principal steps over the entire potential range. This is an important step forward because it allows us to connect individually triggered reaction steps with continuous turnover.
- 2) To the best of our knowledge, decomposing a continuously occurring electrocatalytic reaction into the rate of its constituent steps over the entire relevant

potential range, has never been demonstrated before. This is a very important tool in the electrocatalytic toolkit because it not only provides unprecedented insight into the origin of rate-limitations in complex multi-step reactions, but it also provides an opportunity to evaluate the impact of parameter modifications on all the principal steps of a reaction independently. This opens avenues to the rational design of electrocatalytic pathways.

Comment 1.3: Furthermore, the impact in the energy field of the current study may be limited given that propane fuel cells are not a prominent energy conversion technology. Furthermore, if the limitations of propane fuel cells were overcome, their impact would still be limited, given that propane is at an economic disadvantage against other fuels (e.g., natural gas), given its higher cost of production.

Response 1.3: We suspect that future energy economies will utilize sustainably produced carbon-based fuels. In this system, low-temperature direct hydrocarbon fuel cells may gain favor over combustion technologies for their efficiency and lower pollutant emissions. We expect that the methodology and findings of this work, using propane as a model compound, may be broadly applied to the study of other electrocatalytic hydrocarbon oxidation reactions, including natural gas oxidation. Propane, which we use here, is already an important energy vector because it is the primary component in liquefied petroleum gas and makes up 2% of the primary energy use in the United States.¹

In response to the reviewer comment, we have clarified the goals and motivation of this work in the main text:

Alkane oxidation processes are poised to play a critical role in future energy systems. Carbon-based fuels are expected to remain in use due to their high energy density and widespread availability.³⁻⁷ Moreover, such fuels are likely to be part of a sustainable energy economy. This is reflected in the significant research efforts devoted to their sustainable synthesis via electrocatalytic CO₂ reduction and catalytic biomass conversion.⁸⁻²⁶ Yet, aside from combustion, we lack efficient technologies for recovering the energy stored in these fuels. The development of hydrocarbon energy conversion technologies that surpass combustion in efficiency, reduce pollutants such as NO_x, and simplify system design represents a key opportunity in the transition to a more efficient energy economy. Fuel cells capable of generating electricity from electrocatalytic hydrocarbon oxidation offer a promising path toward meeting these goals (Figure 1a).²⁷

Low-temperature direct hydrocarbon fuel cells (LT-DHFC) have garnered interest in past and present-day research.²⁸⁻³⁹ Nevertheless, their performance is poor compared to hydrogen fuel cells.⁴⁰⁻⁵⁰ Similarly, fuel cells operating on oxygenates such as alcohols suffer from low efficiency and incomplete oxidation of alkyl chains.⁵¹⁻⁶⁹ A key limitation across these systems is the inadequate understanding of the mechanistic bottlenecks in steady-state electrocatalytic alkyl chain oxidation. To address this gap, the fundamental study presented herein investigates the electrocatalytic oxidation of alkanes, using propane as a model alkane,

and systematically relates the steady-state oxidation rate to the potential-dependent kinetics of its principal steps.

Comment 1.4: However, the techniques studied by the authors could lead to the ability to selectively and partially oxidize propane to higher value chemicals (e.g., oxygenates) rather than fully oxidizing it to CO₂, but this is unfortunately a challenging task as described previously by the authors (DOI: 10.1038/s41929-024-01218-0) and further demonstrated by the presented study.

Response 1.4: We agree with the reviewer that partial oxidation pathways are interesting. The objective of the current work is the total oxidation of alkanes to CO₂ for application in fuel cells and our newly added data (see below) indicates that CO₂ is the only product formed. [Redacted].

To provide more clarity, we have modified the main text to indicate that no partial oxidation was observed in this work:

CO₂ was the only observed product (Figure S21) and control experiments under He (Figure S18) confirmed that the observed CO₂ originated from propane oxidation rather than from electrolyte impurities.^{31,35,99}

The SI was modified to incorporate new experiments that indicate that no partial oxidation was taking place. This finding is consistent with prior literature:

During the application of each oxidative turnover potential, no increase in the ionic currents for the mass fragments related to oxygenate formation (m/z 31, 32, 33, 47, 58, 58) other than CO₂ (m/z 16) was observed. This is consistent with prior literature.^{9,10}

Figure S21. EC-MS ionic currents during the constant-potential oxidation of propane-saturated solution at 60 °C. “M” notation indicate m/z value. Mass fragments associated with the generation of C1 (methanol, M31), C2 (ethanol, M47), and C3 (propanol, M59 and acetone, M58) oxygenates, O₂ (M32), and CO₂ (M16) that do not overlap with propane mass fragments are shown. Integrated area under M16 curves are shown as shaded areas. Applied potential: (a) 0.5 , (b) 0.7 V, (c) 0.9, (d) 1.1 V.

Comment 1.5: These points highlight the limitation on the impact of the paper in the energy field. However, in this reviewer’s opinion, the study presented is of very high quality, has been performed with a remarkable level of rigor, and would likely be well received by the electrocatalysis community, where the impact is more evident than in the energy field.

Response 1.5: We very much appreciate the reviewer’s comment on the quality of the work. Based on the reviewer’s suggestion, we have opted to transfer the manuscript.

Comment 1.6: There are also several important points that the authors should consider prior to publication:

Response 1.6: Answers to the specific comments are provided in the following.

Comment 1.7: A key concern in this study is the long timescales associated with the molecular processes involved in propane oxidation. For instance, Figure 4b shows that the maximum adsorbed propane is not reached even after 4 minutes across nearly all applied adsorption potentials. Ideally, adsorption should occur on a millisecond timescale to ensure rapid and efficient electrocatalytic turnover. This slow adsorption rate raises questions about potential surface limitations or transport effects that may hinder adsorption kinetics. A more detailed discussion of factors influencing these timescales, such as surface coverage dynamics, competitive adsorption with electrolyte species, or diffusion limitations, would be valuable.

Response 1.7: In response to the reviewer’s suggestion, we have added a detailed discussion of the factors known to impact the rate of propane adsorption to the main text. This analysis explains the potential-dependent adsorption trend observed in our work and indicates that mass transport may not be limiting adsorption.

Even though the reactions appear slow, the time scales are not entirely different from the turnover frequencies observed for catalytic processes on surfaces. For example, the water-gas shift reaction (TOF = $1 - 4 \times 10^{-3} \text{ s}^{-1}$ ², $0.7 - 3.5 \text{ s}^{-1}$ ³), the Haber-Bosch process (TOF = $\sim 10 \text{ s}^{-1}$ ^{4,5}), and methane steam reforming (TOF = $0.5 - 19.9 \text{ s}^{-1}$ ⁶) have reported turnover frequencies that can be on the time scale of seconds.

The following was added to the main text:

*Adsorption rates were estimated by dividing the total adsorbed propane by t_{ads} (Figure S4), and rates obtained at $t_{ads} = 30 \text{ s}$ are shown in **Figure 4c**. The rate was negligible below 0.2 V, peaked at 0.4 V, decreases modestly between 0.5 and 0.7 V, and dropped sharply*

above 0.8 V. At longer adsorption times, adsorbate quantities began to plateau, indicating surface saturation. All measured rates were below the diffusion-limited regime (see SI), consistent with a kinetically limited adsorption process, as previously proposed by Cairns and coworkers.⁷⁷ Our measured trends differ quantitatively from earlier reports, which typically placed maximum adsorption near 0.3 V with a sharp decrease at more reductive and oxidative potentials.^{71,74,76,77,81,88,89,93–95} We attribute these differences to our inclusion of CO₂ generated during the adsorption step, made possible by real-time EC-MS measurements.

The observed potential dependence of propane adsorption and saturation behavior can, for example, be interpreted using the water displacement theory of electrochemical adsorption, originally proposed by Bockris and coworkers.¹⁰⁰ According to this model, the applied potential influences neutral molecule adsorption by modulating the strength of water-electrode interactions.^{101,102} Since propane must displace interfacial water during adsorption, the process is favored near the potential of zero charge (PZC), where water-electrode interactions are minimized.¹⁰³ For polycrystalline Pt, the PZC is estimated to lie near 0.3 – 0.4 V, consistent with the observed peak in adsorption rate.^{104,105} At more reductive potentials, competitive hydrogen adsorption via underpotential deposition inhibits propane adsorption.^{77,93,94}

At potentials between 0.5 and 0.8 V, the onset of dissociative water adsorption promotes the formation of *OH which has competing effects on propane adsorption.^{106–109} Surface-bound OH (and Pt=O above 0.8 V) can block active sites, hindering further adsorption.^{110–112} On the other hand, *OH can react with *CO, the intermediate formed from propane, enabling continuous oxidation.¹¹³ This dynamic reduces adsorbate buildup and facilitates sustained adsorption. The real-time CO₂ detection by EC-MS allowed us to capture this effect quantitatively and incorporate it into our adsorption rates. Finally, we note that anion specific adsorption is not expected to impact our results as the perchlorate anion is believed to interact weakly with electrode surfaces.^{114–119}

Calculations and discussion concluding that propane adsorption is not diffusion-limited were added to the SI:

The flux of a species via diffusion is given by:

$$J = -D \frac{dC}{dx} \quad (S8)$$

where J is the diffusion flux in $\text{nmol s}^{-1} \text{cm}^{-2}$, D is the diffusion coefficient, and $\frac{dC}{dx}$ is the concentration gradient. The diffusion coefficient of propane at 60 °C in water was extrapolated to be $2.07 \times 10^{-5} \text{ cm}^2 \text{ s}^{-1}$ from the diffusion measurements performed by Witherspoon and Saraf as shown in **Figure S22**.¹¹ The concentration gradient was calculated from the concentration difference between the bulk solution and electrode surface. Here we assume that under diffusion limited adsorption, the concentration at the electrode surface is zero. The concentration of propane in the bulk was approximated to be $2.0 \times 10^{-3} \text{ M}$ based on measurements reported by Morrison and Billett.¹² The thickness of the diffusion layer was

assumed to be 100 μm which is the thickness of the EC–MS cell electrolyte compartment (the lowest possible flux scenario for this cell and these conditions), and a linear concentration profile was assumed. Substitution into **Equation S8** and solving for J yielded a flux of 4.1 nmol propane $\text{s}^{-1} \text{cm}^{-2}$. Considering that propane adsorption at 0.4 V for 240 s closely approaches saturation (**Figure 4b**) and yields approximately 28 nmol CO_2 , we can estimate that 9.3 nmol propane was adsorbed to the 0.196 cm^2_{geo} electrode. This corresponds to a coverage of 47.4 nmol propane cm^2_{geo} . Diffusion limited adsorption of propane should therefore reach full saturation in approximately 11.5 s. As shown in **Figure 2b** and **4b**, full saturation was not achieved within 11.5 s thus indicating that the adsorption of propane was not limited by propane diffusion.

Figure S22. Diffusion coefficient of propane in water as a function of temperature as reported by Witherspoon and Saraf.¹¹ Data was fit with an exponential growth function in Microsoft Excel based on the assumption of an Arrhenius-type relationship. The diffusion coefficient of propane in water at 60 °C was calculated from the fit.

Comment 1.8: Furthermore, given that the study explores pulsed potential strategies to enhance oxidation rates, investigating whether the adsorption step could be accelerated would be particularly relevant. Shortening the adsorption duration could significantly improve CO_2 production rates and overall catalytic efficiency.

Response 1.8: We found that adsorption was fastest at shorter time scales (**Figure S4**). This finding was leveraged in the ‘Potential Pulses for the Oxidation of Propane’ section to enhance the overall propane oxidation rate.

[Redacted]

The following has been added to the main text to highlight our goal of increasing alkane adsorption rate in future research:

The ability to identify and decouple the contributions of adsorption, conversion, and oxidation was essential to designing this enhancement strategy. Building on this approach, we

are now exploring methods to further improve alkane adsorption rates and to tune the potential windows for conversion and oxidation steps in electrocatalytic hydrocarbon reactions.

Comment 1.9: In the introduction, the authors emphasize the low current densities of propane fuel cells (24 mA cm⁻² vs. 1.9 A cm⁻² for hydrogen fuel cells) as a major limitation. However, they do not discuss how the reported CO₂ production rates relate to current densities at different applied potentials. If low current density is a key concern, why is it not used as a metric for optimization?

Response 1.9: The current density comparison was introduced to highlight the fact that further research into understanding the bottlenecks of direct alkane fuel cells is required. Our study is of a fundamental nature, focused on developing a methodology that allows us to identify the rate-limiting step of alkane fuel oxidation reactions over the entire potential range. Comparison of current densities with high surface area gas diffusion electrode devices is not possible, as we operate with lower surface areas and in the liquid phase – which was chosen because it allows us to precisely control the electrochemical environment. Furthermore, previous reports do not provide activity normalized to effective surface area, which presents further obstacles to drawing apples-to-apples comparisons.^{7–19}

In response to the reviewer comments, we have changed the main text introduction to more clearly communicate the goals of this study and compare the performance of present-day hydrocarbon fuel cells to other fuel cell technologies in qualitative terms: See revised main text in Response 1.3.

Comment 1.10: Furthermore, the study does not address current efficiency or the impact of competing reactions such as the oxygen evolution reaction (OER), which could affect CO₂ selectivity. Including partial current densities and faradaic efficiency analysis would provide a more comprehensive assessment of the system's practical performance and viability for fuel cell applications.

Response 1.10: We have modified the manuscript to discuss these parameters. No O₂ generation, nor formation of partial oxidation products was observed during propane oxidation. Our study was performed using EC-MS, which allowed us to observe the products generated in each step. No change in the ionic current corresponding to O₂ (m/z 32) was observed during constant-potential oxidation of propane, nor during the oxidative steps in sequential oxidation experiments and pulse experiments. Thus, *O₂ generation is not a factor in our study.* We have also recorded the mass fragments for possible partial oxidation products, none of which were observed. In agreement with prior reports, CO₂ was the only observed product.^{8,20,21}

The main text was revised to communicate these findings, and the data is presented in the SI. The relevant main text and SI additions are discussed in Response 1.4.

Comment 1.11: The authors state, "Our comparatively high propane adsorption rate between 0.4 – 0.7 V can be attributed to the inclusion of the CO₂ produced during the

adsorption step." However, in the "Constant-potential Oxidation of Propane" section, it is indicated that no oxidative turnover occurs at 0.4 V. This raises a discrepancy—if there is no propane oxidation at 0.4 V, how does CO₂ production contribute to the reported adsorption rate in this potential range? Clarification is needed on whether another mechanism accounts for CO₂ formation at 0.4 V or if there is an inconsistency in the interpretation of turnover at low potentials.

Response 1.11: The reviewer is correct. This sentence contained a typo, and we did not measure any CO₂ production at 0.4 V. The range in which propane turnover occurs is now specified and the comparison to previous literature has been revised.

The revised main text now reads:

As expected from the turnover data, CO₂ production was also observed during the adsorption phase for E_{ads} between 0.5 to 0.9 V. We contend that this CO₂ originates from propane that had undergone adsorption, and therefore should be included in the total adsorption rate.

...

Our measured trends differ quantitatively from earlier reports, which typically placed maximum adsorption near 0.3 V with a sharp decrease at more reductive and oxidative potentials.^{71,74,76,77,81,88,89,93–95}. We attribute these differences to our inclusion of CO₂ generated during the adsorption step, made possible by real-time EC-MS measurements.

Comment 1.12: The "Rate of Multi-carbon Adsorbate Conversion" section presents data derived from complex methodologies, with many details deferred to the Supplementary Information (SI). For instance, the determination of adsorbed *CO from linear sweep voltammetry (LSV) required compensation for other contributing effects, but the main text does not clearly explain how these corrections were applied or how the accuracy of the measurements was assessed. As a result, it is difficult for the reader to fully understand how propane conversion below 0.5 V was measured. Rather than relying on the SI for key experimental details, the manuscript would benefit from a clearer discussion within the main text, ensuring that the methodology is transparent and comprehensible without requiring extensive reference to supplementary materials.

Response 1.12 (Part a): To address the reviewer's comment, we have revised the main text to include details on how the conversion of multi-carbon adsorbates was calculated and to discuss how we assessed the accuracy of our method. **Figures 5, S5, and S6** have also been revised to clarify the methods and display more of the values used in our calculations. **Figure S11** and **Table S19** were added to show all data related to the determination of conversion rates below 0.5 V. Both the modified figures and text are shown below:

*For potentials above 0.5 V, where *CO is immediately oxidized to CO₂, the potential program shown in **Figure 5a** was used in a propane saturated electrolyte. After pre-conditioning (Figure S1), the electrode was held at 0.3 V for 120 s to adsorb propane. The potential was then stepped to ' E_{conv} ' (0.5 – 1.1 V) for a time ' t_{conv} ', initiating adsorbate*

conversion and immediate oxidation. Finally, the potential was returned to 0.3 V to terminate the reaction. The resulting CO₂ evolution was monitored by EC-MS (m/z 16), and a representative CO₂ flux trace is shown in **Figure 5a**.

The total CO₂ produced during E_{conv} includes contributions from (i) propane turnover, (ii) oxidation of *CO formed during the initial 0.3 V adsorption step, and (iii) conversion and oxidation of pre-adsorbed propane. To isolate the CO₂ corresponding to the conversion and oxidation of adsorbed propane ' $n_{CO_2}(\text{propane,conv})$ ', we applied the following relationship:

$$n_{CO_2}(\text{propane,conv}) = n_{CO_2}(\text{total}) - (n_{CO_2}(\text{turnover}) + n_{CO_2}(\text{CO,ads}))$$

To determine $n_{CO_2}(\text{turnover})$, we repeated the experiments with $t_{ads} = 0$ s, ensuring that no pre-adsorbed intermediates were present. The resulting CO₂ fluxes represent the background signal from steady-state propane oxidation at each E_{conv} (**Figure 5b**). These values were subtracted from the total CO₂ yield.

To determine $n_{CO_2}(\text{CO,ads})$, we used linear sweep voltammetry (LSV) to oxidatively strip *CO generated during the 0.3 V adsorption step (Figure S6a), following methods established by Cairns and coworkers.⁷⁷ The LSV peak integrated along the baseline was converted to *CO quantity by assuming a 2 e⁻ oxidation per *CO molecule.^{74,86,120–123} We expect that during LSV, further conversion and CO₂ generation will take place, leading to an overestimation of the conversion that occurred at 0.3 V.^{74,87} Because the *CO oxidation peak in LSV is clearly distinguishable from the baseline, we chose to use the voltammetric current instead of the EC-MS signal for this analysis. Voltammograms for $t_{ads} = 0$ s were used as blanks and subtracted from the $t_{ads} = 120$ s signal.

With both corrections applied, we plotted the amount of converted propane $n_{CO_2}(\text{propane,conv})$ in **Figure 5c**, and extracted the conversion rates from fits to the initial linear segments of the data (**Figure 5d**). Between 0.5 and 0.7 V, conversion rates were comparable to the steady-state turnover rate but increased significantly at 0.9 and 1.1 V.

Figure 5c showed plateau-like behavior for 0.9 and 1.1 V. We interpret this as the rapid conversion and oxidation of most of the pre-adsorbed propane. However, the amount of propane converted during t_{conv} fell short of the total adsorbed quantity, even at the most oxidative potentials. This suggests that not all pre-adsorbed propane was converted during these experiments. Additional experiments, in which 1.3 V was applied immediately after E_{conv} , confirmed the presence of unreacted adsorbates (Figure S12). This observation aligns with previous reports of recalcitrant intermediates that persist on the surface and are unlikely to contribute to propane oxidative turnover.^{74,77} Importantly, the corrected yields remained within error of the expected pre-adsorbed amount, supporting the validity of our turnover correction approach.

For $E_{conv} < 0.5$ V, *CO accumulates instead of being oxidized, allowing direct quantification by LSV oxidative stripping. Using the procedure shown in **Figure 5e**, we determined *CO yields for various values of E_{conv} and t_{conv} . As a validation step, we compared blank ($t_{ads} = 0$ s) LSV traces to the shortest conversion time experiments and found similar signals (Figure S11), confirming minimal conversion at early times.^{74,87} An example LSV at

0.4 V and 60 s is shown in **Figure 5e**. Integrated *CO quantities (corrected for the blank) are shown in **Figure 5f**, and conversion rates extracted from linear fits are included in **Figure 5d**. Below 0.5 V, conversion was slow but accelerated slightly from 0.1 to 0.4 V. Having quantified the potential-dependent rate of *CO formation, we next investigated the kinetics of *CO oxidation to determine how this final step regulates overall propane turnover.

To further improve the clarity of our work, **Figure 5** has been revised to include visual representations of the CO_2 sources (colored boxes), example EC-MS CO_2 fluxes (**Figure 5a**), consumed propane due to turnover above 0.5 V (**Figure 5b**), and example LSV curves (**Figure 5e**):

Figure 5. (a) Electrode potential program used to study the conversion and oxidation of pre-adsorbed propane adsorbates for $E_{conv} = 0.5, 0.6, 0.7, 0.9,$ and 1.1 V and $t_{conv} = 1, 5, 10, 30, 60,$ and 240 s. Top inset illustrates relative contributions from each CO_2 source. CO_2 fluxes for $E_{conv} = 0.7$ V and $t_{conv} = 240$ s and $E_{turnover} = 0.7$ V and $t_{turnover} = 240$ s are shown. (b) Quantity of propane consumed due to turnover during conversion experiments. Each data point is the average of at least two data points collected on separately prepared electrodes and error bars show one standard deviation. Dashed lines connecting data points were added as guides to the eye. (c) Quantity of converted propane (corrected for propane turnover) as a function of E_{conv} and t_{conv} . Each data point is the average of at least two data points collected on separately prepared electrodes and error bars show one standard deviation. Dashed lines connecting data points were added to guide the eyes. The blue dotted line at 13.3 nmol $s^{-1} C^{-1}$ represents the average quantity of propane converted to *CO during the adsorption step. (d) Rate of propane conversion as a function of the applied potential. The grey dashed line separates the data collected using the two different methods. (e) Electrode potential program for the study of propane conversion to *CO for $E_{conv} = 0.1, 0.2, 0.3, 0.4$ V and $t_{conv} = 10, 30,$ and 60 s. Illustrations show that adsorption and conversion occur during the application of E_{conv} and *CO oxidation occurs during the LSV. Example LSV peak integration for a blank and a $E_{conv} = 0.4$ V and $t_{conv} = 60$ s experiment are shown. (f) Quantity of converted propane as a function of E_{conv} and t_{conv} . Linear fits of the data are shown as dashed lines.

To clarify the method used to study conversion rates below 0.5 V using liner sweep voltammetry (LSV), **Figure S6** has been updated to include the LSVs that were collected and evaluated:

Figure S6. Oxidative linear sweep voltammetric (LSV) stripping for the quantification of *CO formation. The integral of the peak corresponding to *CO oxidation is shown as a shaded area. Scan rate = 100 mV s^{-1} (a) LSV after propane adsorbed at 0.3 V for 120 s to obtain the *CO correction for multi-carbon adsorbate conversion experiments above 0.5 V. Note that experiments using this technique halted the LSV at 0.875 V to only consider CO_2 produced by the oxidation of *CO species corresponding to this peak. The LSV here is carried out until 1.4 V to more clearly show the peak of interest. (b) LSVs after $t_{\text{conv}} = 10 \text{ s}$. (c) LSVs after $t_{\text{conv}} = 30 \text{ s}$. (d) LSVs after $t_{\text{conv}} = 60 \text{ s}$.

Response 1.12 (Part b): The accuracy of our turnover correction was assessed by performing a carbon balance analysis. In response to the reviewer's comments, we have added details on these measurements to the SI:

In the 'Rate of Multi-carbon Adsorbate Conversion' section, conversion experiments above 0.5 V required correction for the propane converted and oxidized due to continuous turnover. To assess the accuracy of these corrections, we analyzed whether the carbon balance closes. If the correction and analysis is accurate, then the number of carbon atoms released from the oxidation of converted and unconverted propane should sum to the total adsorbed amount, which is on average $336 \text{ nmol C}^{-1}_{\text{o-r}}$ propane for propane adsorbed for 120 s at 0.3 V. To check this, we performed additional experiments that have been added to the SI (Figure S12). These experiments were variations of the conversion experiments shown in Figure 5a where the potential was stepped to 1.3 V directly following the completion of a 240 s conversion step at 0.7 V (Figure S12a) or 0.9 V (Figure S12b), to quantify the number of unconverted carbon atoms that remain on the surface. An increase in the m/z 16 EC-MS signal was observed upon the application of 1.3 V indicating the oxidation of unconverted propane. A smaller quantity of unconverted propane was oxidized after conversion at 0.9 V vs 0.7 V.

For each experiment, the propane oxidized during conversion and the unconverted propane oxidized at 1.3 V were summed. Once corrections were made for the propane oxidized

due to continuous turnover at the conversion potential, we estimated that 430 and 408 $\text{nmol C}^{-1}_{\text{o-r}}$ propane were adsorbed during the initial 0.3 V 120 s adsorption step for conversion at 0.7 (Figure S12a) and 0.9 V (Figure S12b), respectively. These values are within error of the expected 336 $\text{nmol C}^{-1}_{\text{o-r}}$ for propane adsorption at 0.3 V for 120 s.

Figure S12. Modified potential program and m/z 16 EC-MS ionic current response for a multi-carbon adsorbate conversion experiment. Following adsorption at 0.3 V for 120 s and conversion at E_{conv} for 240 s, 1.3 V was applied to the electrode to oxidatively remove residual adsorbates. The quantity of propane oxidized during each peak is labeled. The quantity of propane oxidized due to continuous oxidation (turnover) at E_{conv} is noted in purple and subtracted from the total oxidized propane. (a) $E_{\text{conv}} = 0.7$ V. (b) $E_{\text{conv}} = 0.9$ V.

Comment 1.13: In the "Rate of CO Oxidation" section, the authors state, "This observation suggests that propane multi-carbon adsorbates may inhibit *CO oxidation, thus leading to the accumulation of CO during the adsorption of propane at 0.3 and 0.4 V." However, this statement is unclear. If *CO accumulates on the surface, it should still undergo oxidation in the same manner as in the CO oxidation experiment. It is not evident why the presence of multi-carbon adsorbates would specifically inhibit *CO oxidation, and further clarification is needed. Rewording this sentence to better explain the proposed inhibition mechanism or providing supporting evidence would improve clarity.

Response 1.13: We thank the reviewer for pointing this out. We do not need to invoke an interaction between adsorbed CO and propane to explain the low rate of CO_2 generation, which is likely simply due to the low CO^* concentration on the surface at 0.3 and 0.4 V. The main text has been changed as follows:

*CO oxidation rates increased gradually between 0.2 to 0.6 V, then rose more sharply at more oxidative potentials. The plateauing behavior of the CO_2 yields in Figure 6b indicates that *CO was rapidly consumed under oxidative conditions.^{122,124} Notably, we detected non-zero CO_2 production from CO oxidation at 0.3 and 0.4 V. While propane oxidative turnover was not observed in this potential range, the shared *CO intermediate suggests that steady-state propane turnover may nonetheless be possible at these lower potentials. However, given the low *CO formation rates observed at 0.3 – 0.4 V, it likely only accounts for a small fraction of surface adsorbates, which may explain why CO_2 production from propane remains below the EC-MS detection limit under these conditions. Taken together with the results of the previous sections, this analysis completes the mechanistic decomposition of propane oxidation and enables a step-resolved comparison of potential-dependent rates across the full reaction sequence.*

Comment 1.14: In the introduction, the authors state, "This achievement catalyzed research efforts that established a basic understanding of electrocatalytic alkane reactivity" followed by 68 references. While this acknowledges the breadth of prior work, it does not provide any insight into how these studies have specifically advanced the field. The authors should either expand this literature review to briefly summarize key contributions from these references or significantly reduce the number cited to focus on the most relevant works and if possible, on key review papers that summarize this literature.

Response 1.14: The introduction has been significantly revised to more clearly discuss our motivation and goals. During revision, the statement that these citations were associated with was removed and the citations are now more closely associated with discussion of their respective findings. We intended to provide a comprehensive acknowledgement of prior work; however, we agree that this exhaustive list of 68 references was not effective. 33 less relevant citations were removed. The remaining citations remain in the main text to support our statements and findings. The revised introduction can be seen in Response 1.3.

Reviewer #2:

Comment 2.1: This work focuses on the study of the three processes that are involved in the oxidation of propane on Pt: adsorption, conversion and oxidation, applying electrochemical mass spectrometry. This is a fundamental study in which, by applying different potential routines, information is obtained about the indicated processes. Considering the eminently practical nature of this journal, the manuscript cannot be accepted for publication in Nature Energy.

Response 2.1: We thank the reviewer for the careful analysis of our work and the feedback.

Comment 2.2: Some aspects to improve the manuscript for publication are the following:
1) Characterization of the Pt electrode is necessary. Reference is made to a previous work in which only a XPS and a SEM image are shown but without analysis, to know the oxidation state of Pt, the particle/crystallite size, etc.

Response 2.2: In response to the reviewer's request, we have added extensive characterization of our platinized Pt electrode using X-ray diffraction, X-ray photoelectron spectroscopy, and scanning electron microscopy. Mention of this characterization and summarized findings have been added to the main text:

The catalyst exhibited a rough polycrystalline structure and was characterized by scanning electron microscopy (SEM), X-ray diffraction (XRD) and X-ray photoelectron spectroscopy (XPS) (see SI).

Catalyst characterization results and discussion have been added to a new *Catalyst Characterization* section in the Supplementary Information:

I. Catalyst Preparation for Material Characterization

Platinized Pt electrodes were prepared as described in the Catalyst Preparation section. After catalyst deposition and rinsing with Milli-Q water, the electrode was mounted in the EC-MS cell. In a 1 M HClO₄ solution at room temperature, 20 CV cycles from 0 to +1.4 V were performed at 50 mV s⁻¹ to pre-condition the electrode. The electrode was then removed from the cell and rinsed with Milli-Q water. The sample was then allowed to dry under ambient conditions before being placed in a PTFE sample holder for transport to analysis facilities.

II. X-ray diffraction

*X-ray Diffraction (XRD) was performed on the platinized Pt catalyst using a Bruker D8 Discovery X-ray diffractometer using a Cu K_α X-ray source, 0.5 mm spot size, and a Vantec 500 area detector. The X-ray diffraction pattern is shown in **Figure S31** and agrees with the XRD pattern of Pt in the PDF-5+ 2025 database.¹³*

Figure S31. X-ray diffraction pattern of platinumized Pt catalyst. Peaks have been labeled with corresponding reflections.

Reflections were assigned according to those listed for Pt in the PDF-5+ 2025 database. Crystallite size was determined using Scherrer analysis available in the Bruker DIFFRAC.EVA software using a shape factor, K , equal to 0.89 and instrument line broadening of 0.05. The mean crystallite size was determined to be 136.8 Å when analyzing the peak broadening of the $2\theta = 39.810$ peak.

III. X-ray Photoelectron Spectroscopy

Surface-sensitive elemental analysis was performed on the platinumized Pt catalyst using X-ray photoelectron spectroscopy (XPS). A Thermo k -alpha X-ray photoelectron spectrometer with an Al $K\alpha$ X-ray source with a hemispherical electron energy analyzer was used. The results are shown in **Figure S32**. The peak binding energy positions were compared to reference materials.¹⁴⁻¹⁶

Figure S32. X-ray photoelectron spectroscopy spectra of platinumized Pt catalyst. The origin of each peak is labeled.

The peak at 71.1 eV corresponds closely to the reference value of 71.2 eV for Pt 4f_{7/2}. This peak is not shifted to higher binding energies, indicating an absence of Pt binding to

electronegative elements such as O, thus suggesting that Pt exists predominantly in the reduced state. If Pt oxides or Pt hydroxides were present, a Pt 4f_{7/2} binding energy between 74 – 75 eV or 72 – 73 eV would likely be observed, respectively.¹⁴

The presence of trace elements was analyzed using high-resolution XPS spectra (Figure S33). A Cl 1s binding energy of 198.1 eV was observed. This binding energy is closest to the Cl 1s binding energy found for tetrachloroplatinate (198.4 – 198.7 eV) and hexachloroplatinate (198.9 eV) samples. This suggests that the observed trace Cl likely exists in a complexed form, likely residues from the chloroplatinic acid deposition solution. The detected Cl does not come from the perchloric acid electrolyte since a Cl 1s binding energy of 208.33 eV is expected for perchlorates.¹⁴ The dominant observed C and O peaks likely correspond to adventitious carbon and water, which are nearly unavoidable and therefore common in XPS analysis. A small peak at 688.7 eV might point to C–F bonding, potentially originating from the PTFE sample holders used to transport and handle the sample. No presence of Pb, which is used as a promotor during electrodeposition, was detected.

Figure S33. High-resolution x-ray photoelectron spectroscopy of platinumized Pt catalyst. Ten scans were collected for each spectrum. (a) Pt 4f. (b) Cl 2p. (c) O 1s. (d) C 1s. (e) Pb 4f. (f) F 1s.

IV. Scanning Electron Microscopy

Scanning electron microscopy (SEM) was used to assess the topography of the platinized Pt catalyst. SEM analysis was performed using a Zeiss Gemini 450 at an acceleration voltage of 20.00 kV. We observed that the platinized Pt catalyst exhibits a hierarchical, fractal-like morphology.

Figure S34. Scanning electron microscope images of the platinized platinum catalyst. Magnification: (a) 1 k, (b) 5 k, (c) 10 k, (d) 25 k, (e) 50 k, (f) 100 k.

Comment 2.3: The conditions of the experiments vary without justification: in the cyclic voltammetry a temperature of 80 °C was used and in the mass spectrometry experiments 60 °C; in some cases, the solution is deoxygenated with Ar and in others with He;

Response 2.3: To address the reviewer's comment, we have repeated the cyclic voltammograms at 60 °C, in the EC-MS setup, and using He as inert gas. These are the exact conditions under which all EC-MS measurements were carried out. All data have thus now been collected under the same conditions and in the same cell. **Figure 2b** has been revised to include these new data. The main text and SI have been changed accordingly and the previous measurements carried out at 80 °C have been moved to a new section of the SI.

The following section of the main text was revised to detail the consistent temperature:

All experiments were conducted in a stagnant thin-layer EC-MS cell in 1 M HClO₄ at 60 °C.

The following section of the main text was revised to detail the use of He as inert gas:

To verify that these features arise from propane-derived species, we repeated the experiments in He saturated electrolyte. Even after 900 s of adsorption, oxidative currents

were negligible under these conditions compared to those in propane saturated electrolyte (**Figure 2b**), confirming that electrolyte impurities do not significantly contribute to the observed signal.

We have revised **Figure 2** to include the CV collected in the EC-MS set-up at 60 °C, as well as the blank measurement under He:

Figure 2. (a) Electrode potential program for the study of propane adsorption at 0.3 V for $t_{ads} = 60, 300,$ and 900 s and subsequent cyclic voltammetric oxidative stripping from 0.3 to 1.3 V. (b) Cyclic voltammetric oxidative stripping traces after adsorption of propane at 0.3 V for 60, 300, and 900 s. Cycle 2 immediately follows Cycle 1 and represents the case for $t_{ads} = 0$ s. Cycle 2 and He blank traces from $t_{ads} = 900$ s are shown. Oxidative peaks are labeled Peak I, II, and III where each Peak corresponds to the oxidation of the indicated adsorbate. Only one of several possible multi-carbon adsorbates corresponding to Peaks II and III are shown for simplicity. Scan rate = 20 mV s⁻¹.

Comment 2.4: 0.3 V has been chosen as the adsorption potential but it is not indicated why (it does not correspond to the maximum adsorption)....The same experimental conditions must be used, or, where appropriate, the changes must be justified.

Response 2.4: 0.3 V was chosen because it corresponds to the potential at which adsorption of propane is favorable while still being reductive enough to limit conversion reactions and prevent turnover during the adsorption process.

To clarify this, the main text was revised as follows:

Propane adsorption was then initiated by applying 0.3 V for durations ranging from 60 to 900 s, conditions chosen to favor adsorption while limiting adsorbate conversion.^{74,87}

Comment 2.5: Blank adsorption experiments have to be performed, that is, repeat the experiments applying the adsorption potential and time routines without propane. The Pt electrode is very active against impurities, even more so in perchloric acid medium where the anion is weakly adsorbed, so achieving targets is very important. Include the blank experiments in the supporting information.

Response 2.5: In response to the reviewer's comment, we carried out EC-MS blank experiments under He. The CO₂ yields observed during these blank experiments were less than 6 % of that observed under propane (see Figures below). We are thus

confident that the oxidation of organic impurities does not significantly influence the reaction and does not affect the interpretation of the data.

The blank measurements have been added to the SI:

Figure S17. Representative electrode potential program and CO₂ flux observed during $E_{ads} = 0.3 V$, $t_{ads} = 240 s$ adsorption experiments performed under He or propane. The CO₂ flux was calculated using the m/z 16 MS ionic current. Under He the CO₂ yield was 49.3 nmol C⁻¹ o-r CO₂. Under propane the CO₂ yield was 884.7 nmol C⁻¹ o-r CO₂.

Figure S18. Representative electrode potential program and CO₂ flux observed during $E_{turnover} = 0.7 V$, $t_{turnover} = 240 s$ constant-potential oxidation experiments performed under He or propane. The CO₂ flux was calculated using the m/z 16 MS ionic current. Under He the CO₂ yield was 20.2 nmol C⁻¹ o-r CO₂. Under propane the CO₂ yield was 880.7 nmol C⁻¹ o-r CO₂.

Figure S19. Representative electrode potential program and CO₂ flux during a conversion experiment using $E_{conv} = 0.7 V$ and $t_{conv} = 240 s$, with pre-adsorption at $E_{ads} = 0.3 V$ and $t_{ads} = 120 s$. The experiment was carried out under He or propane. The CO₂ flux was calculated using the m/z 16 MS ionic current. Under He the CO₂ yield was 19.0 nmol C⁻¹ o-r CO₂. Under propane the CO₂ yield was 1406.9 nmol C⁻¹ o-r CO₂.

The following discussion of the blank experiments has been added to the main text:

Control experiments with $t_{ads} = 0 s$ were subtracted from the data but contributed less than 6% of the total signal measured at 0.4 V for 240 s (Table S9), and additional controls in He-

saturated electrolyte confirmed that electrolyte impurities did not significantly influence the measurements (Figure S17).

CO₂ was the only observed product (Figure S21) and control experiments under He (Figure S18) confirmed that the observed CO₂ originated from propane oxidation rather than from electrolyte impurities.^{31,35,99}

Control experiments under He (Figure S19) confirmed that the observed CO₂ originated from propane oxidation rather than from electrolyte impurities.

The fact that no meaningful amount of electrolyte impurities is present was further supported by cyclic voltammetry experiments carried out under He, which show markedly smaller oxidative current peaks than seen under propane. The relevant CV traces are shown in Response 2.3 and the following was added to the main text:

To verify that these features arise from propane-derived species, we repeated the experiments in He saturated electrolyte. Even after 900 s of adsorption, oxidative currents were negligible under these conditions compared to those in propane saturated electrolyte (Figure 2b), confirming that electrolyte impurities do not significantly contribute to the observed signal

Comment 2.6: The experiments and results are described in a complicated way. The clarity of the paper must be improved so that the reader can follow the reasoning.

Response 2.6: In response to the reviewer's comment, we have substantially revised the main text and figures by adding data, discussion, equations, and interpretation to clarify the experimental approach and results. Because the changes are extensive, we here provide a list of modifications and refer the reviewer to the main text for details.

- The 'Introduction' has been revised to more clearly motivate our work and communicate the goals of this fundamental study.
- To increase the clarity of the experimental method, the 'Rate of Propane Adsorption' section was revised and CO₂ flux traces included in **Figure 4a**. Additional discussion and interpretation of our results were also added. Furthermore, a detailed discussion of the factors known to influence alkane adsorption was added to explain our experimental observations.
- To increase the clarity of the experimental method, the 'Constant-Potential Oxidation of Propane' section was revised, and a CO₂ flux trace was added to **Figure 3a**.
- To increase the clarity of the experimental method, the 'Rate of Multi-Carbon Adsorbate Conversion' section was significantly revised, and an equation was added to more clearly explain how our corrections were applied. Discussion was added on how we validated our measurements using blank and control

experiments. The magnitude of corrections made for turnover are now shown in the revised **Figure 5b**. To clarify our experimental results, CO₂ flux traces were added to **Figure 5a**. Example LSV traces were added to **Figure 5e** to demonstrate the LSV oxidative stripping method. Additional discussion and interpretation of experimental results was added to guide the reader.

- To better explain our experimental approach and the interpretation of the results, we also revised the 'Rate of CO Oxidation', 'Identification of Propane Oxidation Bottlenecks', and 'Potential Pulses for the Oxidation of Propane' sections.
- The 'Conclusion' section was revised to clarify the impacts of our findings.

Reviewer #3:

Comment 3.1: In this work, the authors an EC-MS study of the electrochemical propane oxidation reaction, focusing on three potential regimes to isolate individual processes occurring in each potential regime. By monitoring the quantity of CO₂ produced as a function of potential and time using a variety of stepped voltammetric profiles, they infer the rates of dissociative propane adsorption, CO oxidation, and a complex process that the authors term 'conversion', which encompasses all of the bond scission and oxidation events that lead to the formation of CO. While the approach is innovative and the mechanistic conclusions are intriguing, I have some significant questions about the experimental design and methodology, which are provided below.

Response 3.1: We thank the reviewer for the careful analysis of our work and the constructive feedback.

Comment 3.2: A general clarification: Propane is always flowing in the experiments that have a separate 'propane adsorption' step followed by an oxidation or conversion step, correct? Is it possible to perform this experiment where the propane is adsorbed in the 'adsorption' step and then unadsorbed propane purged out prior to the oxidation/conversion step?

Response 3.2: The electrode was continuously exposed to propane as these are the actual conditions under which propane fuel cells operate.

In response to the reviewer's comment, we carried out a new experiment where the oxidation step at longer time points were performed after flushing the cell with helium. These experiments confirm that the adsorbed amount of propane is similar whether the potential was held under He or propane.

We added the following section to the Supplementary Information:

To determine if the presence or absence of propane during the oxidation step influences reaction outcomes, an experiment was performed where propane was adsorbed to the electrode at 0.3 V for 120 s before the gas supply was changed to He (Figure S20). We then maintained the potential at 0.3 V for 45 minutes under He before applying 1.3 V for 3 minutes to promote the oxidation of adsorbed propane. Upon changing the gas supply to He, a decrease of the major propane fragments (M28, M43, M44) was observed, which confirms the removal of solution phase propane (Figure S20a). Upon the application of 1.3 V, we observed peaks in M16 and M44 ionic currents, indicating the production of CO₂ from adsorbed propane (Figure S20b). Calculation of the CO₂ yield from the M16 peak area at 2960 s yielded 813 nmol C⁻¹_{o-r} CO₂. This value was close to adsorption experiments performed at E_{ads} = 0.3 V for t_{ads} = 120 s under continuous propane flow (907 nmol C⁻¹_{o-r} CO₂). The similar CO₂ yield (10% difference) suggests that propane does not significantly desorb from the surface after changing the gas supply to He and that the reaction outcome is not altered by

the presence or absence of propane. [...] Furthermore, the absence of an increase in the M43 signal suggests that propane molecules were not being desorbed during the oxidation step.

Figure S20. Modified adsorption experiment where after adsorption at $E_{ads} = 0.3$ V for $t_{ads} = 120$ s the gas supply was changed to He and the potential maintained at 0.3 V until stepping to 1.3 V at 2960 s. “M” notation indicate m/z value. (a) Larger ionic current scale showing the decay of the major propane fragments (M28, M43, M44) upon changing the gas supply to He (b) Magnified ionic current scale showing the M16 and M44 peak upon application of 1.3 V.

Reference to these experiments was added to the main text:

Reactive loss was not observed for E_{ads} below 0.5 V and purging with He after propane adsorption yielded CO_2 quantities consistent with the ones obtained under continuous propane flow (Figure S20).

Comment 3.3: As a corollary to the above, if you did flow an inert gas through the system after the set adsorption time/potential hold, would propane be desorbed from the surface, and could that be monitored by MS? If so, does the desorbed propane quantity match with the CO_2 quantity measured? This would be a nice confirmation that all of adsorbed propane is indeed converting to CO_2 .

Response 3.3: We answer this question as part of Response 3.2, where we show that no significant propane desorption occurred within the analytical detection limit.

Comment 3.4: In the ‘conversion’ section, the authors discuss the potential convolution of signal due to propane turnover because the solution remains saturated with propane. As a control, they performed an experiment with zero pre-adsorption. It would be helpful to include the plotted data for this control experiment in the same format as Figure 5c so the reader can understand how significant (or minor) this correction was.

Response 3.4: To address the reviewer’s comment, Figure 5 has been revised to include the measurements of propane turnover at the conversion potential (Figure 5b):

Figure 5. (a) Electrode potential program used to study the conversion and oxidation of pre-adsorbed propane adsorbates for $E_{conv} = 0.5, 0.6, 0.7, 0.9,$ and 1.1 V and $t_{conv} = 1, 5, 10, 30, 60,$ and 240 s. Top inset illustrates relative contributions from each CO₂ source. Bottom inset: CO₂ fluxes for $E_{conv} = 0.7$ V and $t_{conv} = 240$ s and $E_{turnover} = 0.7$ V and $t_{turnover} = 240$ s are shown. (b) Quantity of propane consumed due to turnover during conversion experiments. Each data point is the average of at least two data points collected on separately prepared electrodes and error bars show one standard deviation. Dashed lines connecting data points were added as guides to the eye. (c) Quantity of converted propane (corrected for propane turnover) as a function of E_{conv} and t_{conv} . Each data point is the average of at least two data points collected on separately prepared electrodes and error bars show one standard deviation. Dashed lines connecting data points were added to guide the eyes. The blue dotted line at $13.3 \text{ nmol s}^{-1} \text{ C}^{-1}$ represents the average quantity of propane converted to *CO during the adsorption step. (d) Rate of propane conversion as a function of the applied potential. The grey dashed line separates the data collected using the two different methods. (e) Electrode potential program for the study of propane conversion to *CO for $E_{conv} = 0.1, 0.2, 0.3, 0.4$ V and $t_{conv} = 10, 30,$ and 60 s. Illustrations show that adsorption and conversion occur during the application of E_{conv} and *CO oxidation occurs during the LSV. Example LSV peak integration for a blank and a $E_{conv} = 0.4$ V and $t_{conv} = 60$ s experiment are shown. (f) Quantity of converted propane as a function of E_{conv} and t_{conv} . Linear fits of the data are shown as dashed lines.

Comment 3.5: Another general clarification: the authors monitor the mass signal for oxygen, m/z 16, as a proxy for CO₂ and calibrate it using CO₂ in the pure feed. What is the strength of the m/z 16 signal for other oxygen-containing products that may be generated during this experiment? If CO, propanol, methanol, or any number of other C1, C2, and C3 oxygenates are generated, how much would they convolute the m/z 16 signal?

Response 3.5: We collected the EC-MS signals of solutions containing varying concentrations of methanol, ethanol, and 2-propanol. These measurements showed that the m/z 16 signal was not correlated to the alcohol concentration. We can thus confidently state that the m/z 16 signal is not convoluted by oxygenates (see figure below). Furthermore, our data show that CO is rapidly oxidized to CO₂ under our experimental conditions. The following has been added to the SI:

EC-MS signals were collected for solutions containing varying concentrations of methanol, ethanol, and 2-propanol (**Figure S16**). The m/z 16 signal was, however, not correlated to the alcohol concentration and the m/z 16 signal is therefore not convoluted by the potential generation of oxygenates. A slight decrease in m/z 16 and 32 signals was observed as the concentration of alcohol was increased. We attribute this to the slow removal of O_2 from the EC-MS vacuum system with increased experiment time, as the concentration of alcohols was sequentially increased.

Figure S16. EC-MS detection of alcohols. “M” labels indicate m/z value. M16 and M32 are plotted using the left y-axis. M31, M33, M47, M58, M59 are plotted using the right y-axis. Dashed lines added as guides to the eye. (a) Calibration for 0, 0.01, 0.1, and 1 M methanol. (b) Calibration for 0, 0.01, 0.1, and 1 M ethanol. (c) Calibration for 0, 0.001, 0.01, 0.1, and 1 M 2-propanol.

The following experimental details were added to the SI:

The detection of oxygenates using the EC-MS system was evaluated by recording the ionic current for varying concentrations of methanol, ethanol, and 2-propanol. The cell was assembled with a polished Pt stub (99.995%, Pine Research) in the working electrode position and no counter or reference electrode was used. 1 sccm He was flowed during these measurements. The results are shown in **Figure S16**. No correlation between m/z 16 and alcohol concentration was observed.

Comment 3.5 (continued): Have the authors monitored other potential product masses in their system? Have the authors performed steady-state propane oxidation and determined what other products may be produced in significant quantities (either in solution or in the gas phase)?

Response 3.5 (continued): To determine if products other than CO₂ were observed, we performed propane steady-state oxidation experiments at 0.5, 0.7, 0.9, and 1.1 V under the conditions outlined in the ‘Constant-potential Oxidation of Propane’ section. The ionic currents of mass fragments for C1, C2, or C3 oxygenates that do not overlap with propane or CO₂ were monitored (m/z 31, 33, 47, 58, 59) and are shown in **Figure S21**. No increase in mass fragments corresponding to C1, C2, or C3 oxygenate formation was observed. No increase in m/z 32 (corresponding to O₂) was observed either which suggests that the increase in m/z 16 may be attributed to the production of CO₂ without convolution from other products. Additionally, if CO was generated during the reaction, given the ease with which it is adsorbed and oxidized compared to propane, we would not expect it to be persistent in solution and instead be oxidized to CO₂. The following was added to the SI:

Figure S21. EC-MS ionic currents during the constant-potential oxidation of propane-saturated solution at 60 °C. “M” notation indicate m/z value. Mass fragments associated with the generation of C1 (methanol, M31), C2 (ethanol, M47), and C3 (propanol, M59 and acetone, M58) oxygenates, O₂ (M32), and CO₂ (M16) that do not overlap with propane mass fragments are shown. Integrated area under M16 curves are shown as shaded areas. Applied potential: (a) 0.5 V, (b) 0.7 V, (c) 0.9 V, (d) 1.1 V.

During the application of each oxidative turnover potential, no increase in the ionic currents for the mass fragments related to oxygenate formation (m/z 31, 32, 33, 47, 58, 58) other than CO_2 (m/z 16) was observed. This is consistent with prior literature.^{9,10}

Mention of these findings was added to the main text:

CO_2 was the only observed product (**Figure S21**) and control experiments under He (**Figure S18**) confirmed that the observed CO_2 originated from propane oxidation rather than from electrolyte impurities.^{31,35,99}

Comment 3.6: In Figure 4, do the authors think that the propane dissociative adsorption has fully saturated the surface by 240 s? If so, why does the plateau level of adsorbed propane differ so significantly at different potentials?

Response 3.6: In response to the reviewer comment, we have clarified the discussion of the phenomena leading to different plateau values in the main text.

Adsorption was not entirely complete at 240 seconds. Data for 360 s of adsorption is shown in **Figure S10c**. Propane adsorption tended to increase from 240 to 360 s which suggests that the surface is not fully saturated by 240 s for $0.2 \leq E_{\text{ads}} \leq 0.5$ V. The values at 240 s in **Figure 4a** do, however, appear to be approaching plateau values.

The following changes were made to the main text:

At longer adsorption times, adsorbate quantities began to plateau, indicating surface saturation. All measured rates were below the diffusion-limited regime (see SI), consistent with a kinetically limited adsorption process, as previously proposed by Cairns and coworkers.⁷⁷ Our measured trends differ quantitatively from earlier reports, which typically placed maximum adsorption near 0.3 V with a sharp decrease at more reductive and oxidative potentials.^{71,74,76,77,81,88,89,93–95} We attribute these differences to our inclusion of CO_2 generated during the adsorption step, made possible by real-time EC-MS measurements.

The observed potential dependence of propane adsorption and saturation behavior can, for example, be interpreted using the water displacement theory of electrochemical adsorption, originally proposed by Bockris and coworkers.¹⁰⁰ According to this model, the applied potential influences neutral molecule adsorption by modulating the strength of water-electrode interactions.^{101,102} Since propane must displace interfacial water during adsorption, the process is favored near the potential of zero charge (PZC), where water-electrode interactions are minimized.¹⁰³ For polycrystalline Pt, the PZC is estimated to lie near 0.3 – 0.4 V, consistent with the observed peak in adsorption rate.^{104,105} At more reductive potentials, competitive hydrogen adsorption via underpotential deposition inhibits propane adsorption.^{77,93,94}

*At potentials between 0.5 and 0.8 V, the onset of dissociative water adsorption promotes the formation of *OH which has competing effects on propane adsorption.^{106–109} Surface-bound OH (and Pt=O above 0.8 V) can block active sites, hindering further adsorption.^{110–112}*

*On the other hand, *OH can react with *CO, the intermediate formed from propane, enabling continuous oxidation.¹¹³ This dynamic reduces adsorbate buildup and facilitates sustained adsorption. The real-time CO₂ detection by EC-MS allowed us to capture this effect quantitatively and incorporate it into our adsorption rates. Finally, we note that anion specific adsorption is not expected to impact our results as the perchlorate anion is believed to interact weakly with electrode surfaces.^{114–119}*

Comment 3.7: In Figure 5a-c, after the correction is applied for propane turnover (if I'm properly understanding this correction), we anticipate that the amount of propane converted should match the amount of propane adsorbed from the previous experiment, right? The value of converted propane at 240 s at any potential seems to be significantly lower than the amount of propane adsorbed based on the measurement at 0.3 V with 120 s of adsorption time (~350 nmol/C). Why does this discrepancy exist? Is there residual adsorbed propane that is not being converted to CO₂? Or are other products being generated from the adsorbed propane that are not captured? In addition, why does this quantity vary with applied potential? At long time, wouldn't we expect that all of the adsorbed propane should be converted to CO₂ as long as the applied potential is >0.5 V?

Response 3.7: Yes, there is residual adsorbed propane that is not being converted. The data in **Figure 5c** corresponds only to *the quantity of pre-adsorbed propane that is converted, not the total adsorbed amount*. We expect that the *total adsorbed amount* stays approximately constant at 336 nmol C⁻¹_{o-r} for 120 s adsorption at 0.3 V.

To verify if there is additional propane that is not being converted, we performed additional experiments that have been added to the SI (**Figure S12**). These experiments were variations of the conversion experiments shown in **Figure 5a**, where the potential was stepped to 1.3 V directly following the completion of a 240 s conversion step at 0.7 V (**Figure S12a**) or 0.9 V (**Figure S12b**), to quantify the number of unconverted carbon atoms that remain on the surface.

For each of these new experiments, the propane oxidized during conversion and the unconverted propane oxidized at 1.3 V were summed. Once corrections were made for the propane oxidized due to continuous turnover at the conversion potential, we estimated that 430 and 408 nmol C⁻¹_{o-r} propane were adsorbed during the initial 0.3 V, 120 s adsorption step for conversion at 0.7 V (**Figure S12a**) and 0.9 V (**Figure S12b**), respectively. These values are within error of the expected 336 nmol C⁻¹_{o-r} for propane adsorption at 0.3 V for 120 s.

We do not attribute the discrepancy in converted propane (**Figure 5c**) vs the quantity of adsorbed propane (**Figure 4b**) to products other than CO₂. In the experiments performed in response to Comment 3.5 we did not find any evidence of products other than CO₂. We more readily attribute this difference to incomplete conversion of adsorbed multi-carbon intermediates as shown in **Figure S12**.

It is possible that all adsorbed propane is converted to CO₂ during conversion at sufficiently positive conversion potentials. At low potentials however (e.g. 0.5 V), we expect the process to be very slow because our previous work (Lucky *et al.* 2024),²² as well as the work of Cairns,²³ has shown that a fraction of alkane adsorbates requires highly oxidative potentials to be oxidized.

The following figure was added to the SI to show that not all species undergo conversion:

Figure S12. Modified potential program and *m/z* 16 EC-MS ionic current response for a multi-carbon adsorbate conversion experiment. Following adsorption at 0.3 V for 120 s and conversion at E_{conv} for 240 s, 1.3 V was applied to the electrode to oxidatively remove residual adsorbates. The quantity of propane oxidized during each peak is labeled. The quantity of propane oxidized due to continuous oxidation (turnover) at E_{conv} is noted in purple and subtracted from the total oxidized propane. **(a)** $E_{conv} = 0.7$ V. **(b)** $E_{conv} = 0.9$ V.

Discussion of these experiments was added to the main text:

Figure 5c showed plateau-like behavior for 0.9 and 1.1 V. We interpret this as the rapid conversion and oxidation of most of the pre-adsorbed propane. However, the amount of propane converted during t_{conv} fell short of the total adsorbed quantity, even at the most oxidative potentials. This suggests that not all pre-adsorbed propane was converted during these experiments. Additional experiments, in which 1.3 V was applied immediately after E_{conv} , confirmed the presence of unreacted adsorbates (Figure S12). This observation aligns with previous reports of recalcitrant intermediates that persist on the surface and are unlikely to contribute to propane oxidative turnover.^{74,77} Importantly, the corrected yields remained within error of the expected pre-adsorbed amount, supporting the validity of our turnover correction approach.

Comment 3.8: For the experiment in Figure 5e-f, the authors state that they are performing a charge integration of the LSV to determine the quantity of CO₂. In this case, why not integrate the EC-MS signal instead?

Response 3.8: The objective of these experiments was to determine the quantity of *CO that is accumulated on the electrode surface. This is possible by integrating the current along the baseline of the LSV trace. However, there are other oxidative processes occurring during these sweeps, as can be seen by the steeply increasing baseline. These processes contribute to CO₂ formation and therefore make it challenging to isolate the CO₂ derived from *CO oxidation alone. Specifically, conversion of longer chain compounds continues during the oxidative potential sweep.

Therefore, the CO₂ quantified from the EC-MS signal would not be appropriate to measure the quantity of CO present at the surface and would lead to an overestimation. As such, we chose to integrate only the clearly defined peak area and attribute this charge to the oxidation of *CO formed during the chronoamperometric holds as shown in **Figure S6**. In response to the reviewer comment, we have clarified the explanation and reasoning behind this method in the main text:

*To determine n_{CO_2} (CO_{ads}), we used linear sweep voltammetry (LSV) to oxidatively strip *CO generated during the 0.3 V adsorption step (Figure S6a), following methods established by Cairns and coworkers.⁷⁷ The LSV peak integrated along the baseline was converted to *CO quantity by assuming a 2 e⁻ oxidation per *CO molecule.^{74,86,120–123} We expect that during LSV, further conversion and CO₂ generation will take place, leading to an overestimation of the conversion that occurred at 0.3 V.^{74,87} Because the *CO oxidation peak in LSV is clearly distinguishable from the baseline, we chose to use the voltammetric current instead of the EC-MS signal for this analysis. Voltammograms for $t_{\text{ads}} = 0$ s were used as blanks and subtracted from the $t_{\text{ads}} = 120$ s signal.*

...

*For $E_{\text{conv}} < 0.5$ V, *CO accumulates instead of being oxidized, allowing direct quantification by LSV oxidative stripping. Using the procedure shown in **Figure 5e**, we determined *CO yields for various values of E_{conv} and t_{conv} .*

Comment 3.9: As a general comment, it would be helpful if the authors spent a little more time in the main text discussing the control experiments they used to deconvolute competing factors in their experiments. I include one example below where I think clarification would be helpful. The control experiments are extremely important throughout this study because the interpretation of the data hinges upon doing the proper controls. While putting the full details in the SI is helpful, the information is sometimes difficult to parse, and some additional main text discussion will help readers better understand and evaluate the findings. “Other effects, including propane adsorption, propane oxidation, double-layer charging, and surface oxidation may also contribute to the charge passed during the LSV. Therefore, experiments were performed to compensate for these reactions and isolate the charge associated with the oxidation of *CO (see SI for details).”

Response 3.9: The main text and figures have been updated to more clearly detail how the deconvolution of competing factors was performed. A detailed response including the revised text and figures is given in Response 1.12.

References

1. U.S. Department of Energy. Alternative Fuels Data Center: Propane Fuel Basics. *U.S. Department of Energy: Office of Energy Efficiency and Renewable Energy* <https://afdc.energy.gov/fuels/propane-basics>.
2. Zhu, M. & Wachs, I. E. Determining Number of Active Sites and TOF for the High-Temperature Water Gas Shift Reaction by Iron Oxide-Based Catalysts. *ACS Catal* **6**, 1764–1767 (2016).
3. Chen, W.-H. & Chen, C.-Y. Water gas shift reaction for hydrogen production and carbon dioxide capture: A review. *Appl Energy* **258**, 114078 (2020).
4. Qian, J., An, Q., Fortunelli, A., Nielsen, R. J. & Goddard, W. A. Reaction Mechanism and Kinetics for Ammonia Synthesis on the Fe(111) Surface. *J Am Chem Soc* **140**, 6288–6297 (2018).
5. Somorjai, G. A. & Materer, N. Surface structures in ammonia synthesis. *Top Catal* **1**, 215–231 (1994).
6. Jones, G. *et al.* First principles calculations and experimental insight into methane steam reforming over transition metal catalysts. *J Catal* **259**, 147–160 (2008).
7. Grubb, W. T. & Niedrach, L. W. A High Performance Saturated Hydrocarbon Fuel Cell. *J Electrochem Soc* **110**, 1086 (1963).
8. Grubb, W. T. & Michalske, C. J. A High Performance Propane Fuel Cell Operating in the Temperature Range of 150°–200°C. *J Electrochem Soc* **111**, 1015 (1964).
9. Cairns, E. J. Hydrocarbon Fuel Cells with Fluoride Electrolytes. *J Electrochem Soc* **113**, 1200 (1966).
10. Cairns, E. J. High-Performance Hydrocarbon Fuel Cells with Fluoride Electrolytes. *Nature* **210**, 161–162 (1966).
11. Liebafsky, H. A. & Grubb, W. T. Comparative Performance of Normal Alkanes at Platinum Anodes in Fuel Cells. in *Fuel Cell Systems-II* vol. 90 162–170 (American Chemical Society, 1969).
12. Rodríguez-Varela, F. J. & Savadogo, O. Real-Time Mass Spectrometric Analysis of the Anode Exhaust Gases of a Direct Propane Fuel Cell. *J Electrochem Soc* **152**, A1755 (2005).
13. Heo, P., Ito, K., Tomita, A. & Hibino, T. A proton-conducting fuel cell operating with hydrocarbon fuels. *Angewandte Chemie - International Edition* **47**, 7841–7844 (2008).
14. Li, W. S., Lu, D. S., Luo, J. L. & Chuang, K. T. Chemicals and energy co-generation from direct hydrocarbons/oxygen proton exchange membrane fuel cell. *J Power Sources* **145**, 376–382 (2005).
15. Cheng, C. K., Luo, J. L., Chuang, K. T. & Sanger, A. R. Propane fuel cells using phosphoric-acid-doped polybenzimidazole membranes. *Journal of Physical Chemistry B* **109**, 13036–13042 (2005).
16. Rodríguez Varela, F. & Savadogo, O. The Effect of Anode Catalysts on the Behavior of Low Temperature Direct Propane Polymer Electrolyte Fuel Cells (DPFC). *Journal of New Materials for Electrochemical Systems* **9**, 127–137 (2006).
17. Zhu, Y., Tremblay, A. Y., Facey, G. A. & Ternan, M. Petroleum Diesel and Biodiesel Fuels Used in a Direct Hydrocarbon Phosphoric Acid Fuel Cell. *Journal of Fuels* **2015**, 1–9 (2015).

18. Zhu, Y., Robinson, T., Al-Othman, A., Tremblay, A. Y. & Ternan, M. n-Hexadecane Fuel for a Phosphoric Acid Direct Hydrocarbon Fuel Cell. *Journal of Fuels* **2015**, 1–9 (2015).
19. Kong, E. H., Maimani, F., Prakash, G. K. S. & Ronney, P. D. Dynamics of direct hydrocarbon PEM fuel cells. *Sci Rep* **14**, 17865 (2024).
20. Binder, H., Köhling, A., Krupp, H., Richter, K. & Sandstede, G. Electrochemical Oxidation of Certain Hydrocarbons and Carbon Monoxide in Dilute Sulfuric Acid. *J Electrochem Soc* **112**, 355 (1965).
21. Bruckenstein, S. & Comeau, J. Electrochemical mass spectrometry. Part 1.— Preliminary studies of propane oxidation on platinum. *Faraday Discuss. Chem. Soc.* **56**, 285–292 (1973).
22. Lucky, C., Jiang, S., Shih, C.-R., Zavala, V. M. & Schreier, M. Understanding the interplay between electrocatalytic C(sp³)–C(sp³) fragmentation and oxygenation reactions. *Nat Catal* **7**, 1021–1031 (2024).
23. Cairns, E. J. & Breitenstein, A. M. The Kinetics of Propane Adsorption on Platinum in Hydrofluoric Acid. *J Electrochem Soc* **114**, 764–772 (1967).

Response to reviewers

NCOMMS-25-43581-T

Reviewer #4

Comment 1.4. The authors now have explained the reason to adsorb the propane at 0.3 VSHE. Based on previous studies (ref 74 and 87 from the main article, both are citations from the same research group) they concluded that this potential is the best for favoring adsorption while limiting adsorbate conversion. Thus, this reviewer understands that the reason behind choosing 0.3 VSHE is based on previous experience for the same experiments with other alkanes (ethane and butane). Is this statement supported by theoretical calculations? This reviewer suggests to slightly extend or clarify if the precise value of 0.3 VSHE exclusively rises from experimental measurements or if there are any calculations or different literature explaining this effect.

Response 1.4: Experimental studies of alkane adsorption on Pt electrodes by our group and others have generally led to the observation that 0.3 V vs SHE is the optimal potential for the adsorption of alkanes. It is safe to say that the primary evidence for this optimal potential is experimental. However, a recent study simulated the adsorption of propane on Pt (111) using DFT calculations based on the computational hydrogen electrode. This study found that the C₃H₂ adsorbate had a 0.18 eV higher energy barriers for C–C bond breaking compared to C–H bond breaking. This suggests that at 0.3 V, the carbon backbone will more likely stay intact and suggests limited propane fragmentation and conversion at 0.3 V.

The following was added to the main text:

Previous reports have experimentally shown 0.3 V to favor adsorption while limiting adsorbate conversion.^{71,74,87,90–93} Density functional theory calculations showed a 0.18 eV higher energy barrier for C–C vs C–H bond breaking in propane adsorbed on Pt(111) at 0.3 V,⁹³ which supports the experimentally observed limited conversion of propane at this potential.